# Molecular mechanism of Gαi activation by non-GPCR proteins with a Gα-Binding and Activating motif

Alain Ibáñez de Opakua[1,*], Kshitij Parag-Sharma[2,*], Vincent DiGiacomo[2,*], Nekane Merino[1], Anthony Leyme[2], Arthur Marivin[2], Maider Villate[1], Lien T. Nguyen[2], Miguel Angel de la Cruz-Morcillo[2], Juan B. Blanco-Canosa[3], Sekar Ramachandran[4], George S. Baillie[5], Richard A. Cerione[4,6], Francisco J. Blanco[1,7] & Mikel Garcia-Marcos[2]

Heterotrimeric G proteins are quintessential signalling switches activated by nucleotide exchange on Gα. Although activation is predominantly carried out by G-protein-coupled receptors (GPCRs), non-receptor guanine-nucleotide exchange factors (GEFs) have emerged as critical signalling molecules and therapeutic targets. Here we characterize the molecular mechanism of G-protein activation by a family of non-receptor GEFs containing a Gα-binding and -activating (GBA) motif. We combine NMR spectroscopy, computational modelling and biochemistry to map changes in Gα caused by binding of GBA proteins with residue-level resolution. We find that the GBA motif binds to the SwitchII/α3 cleft of Gα and induces changes in the G-1/P-loop and G-2 boxes (involved in phosphate binding), but not in the G-4/G-5 boxes (guanine binding). Our findings reveal that G-protein-binding and activation mechanisms are fundamentally different between GBA proteins and GPCRs, and that GEF-mediated perturbation of nucleotide phosphate binding is sufficient for Gα activation.

[1] CIC bioGUNE, 48160 Derio, Spain. [2] Department of Biochemistry, Boston University School of Medicine, Boston, Massachusetts 02118, USA. [3] Department of Chemistry and Molecular Pharmacology, IRB Barcelona, 08028 Barcelona, Spain. [4] Department of Chemistry and Chemical Biology, Cornell University, Ithaca, New York 14853, USA. [5] Institute of Cardiovascular and Medical Sciences, Department of Molecular Pharmacology, University of Glasgow, Glasgow G12 8QQ, UK. [6] Department of Molecular Medicine, Cornell University, Ithaca, New York 14853, USA. [7] IKERBASQUE, Basque Foundation for Science, 48160 Bilbao, Spain. * These authors contributed equally to this work. Correspondence and requests for materials should be addressed to F.J.B. (email: fblanco@cicbiogune.es) or to M.G.-M. (email: mgm1@bu.edu).

Heterotrimeric (Gαβγ) guanine nucleotide-binding proteins (G proteins) are on/off switches that relay extracellular signals to intracellular effectors and regulate a vast array of physiological processes[1,2]. G-protein activation is achieved when GDP is exchanged for GTP on the Gα subunit, a reaction catalysed by guanine-nucleotide exchange factors (GEFs)[3]. G-protein-coupled receptors (GPCRs) are the archetypical GEFs for heterotrimeric G proteins. Crystal structures[4] in combination with biophysics[5–9], biochemistry[10,11] and computational modelling[12–14] studies have provided detailed knowledge on the mechanism by which GPCRs activate G proteins (Fig. 1a). An obligatory requirement for GPCR-mediated activation is that GDP-loaded Gα is bound to Gβγ. Ligand-occupied GPCRs bind with low affinity to Gαβγ-GDP and facilitate GDP release. The metastable nucleotide-free intermediary forms a high-affinity complex with GPCRs until GTP is loaded, which in turn causes (1) dissociation of the GPCR–G protein complex and (2) dissociation of Gα and Gβγ. GTP hydrolysis returns Gα to the GDP-bound state competent for Gβγ binding.

There are also cytoplasmic non-receptor proteins with GEF activity[15–20], which are structurally unrelated to GPCRs and among themselves. For example, Ric-8 proteins have been extensively characterized biochemically[17,21], and some biophysical and biochemical studies suggest a mode of action similar to that of GPCRs[22,23]. A defined structural module linked to GEF activity has been described only for a subset of non-receptor GEFs, that is, those containing a Gα-binding and -activating (GBA) motif[16,18–20]. This motif is found in evolutionarily unrelated proteins and mediates a mechanism of G-protein activation already present in invertebrates[24]. The GBA motif is a ~30-residue-long sequence present in the human proteins Gα-interacting, vesicle-associated protein (GIV also known as Girdin), dishevelled-associating protein with high frequency of leucines (DAPLE), CALNUC and NUCB2, which bind and activate Gαi proteins (Gαi1, Gαi2 and Gαi3)[18–20,25]. GIV and DAPLE are the best-characterized members of this group. Despite the moderate GEF activity of GIV and DAPLE in vitro, it has been shown that their GBA motif leads to increased G-protein activation in cells, as determined by read-outs that monitor either the formation of Gαi-GTP (for example, conformation-specific antibodies or cAMP)[20,26–28] or free Gβγ (for example, PI3K-Akt or resonance energy transfer-based biosensors)[18,20,27,29,30]. As a consequence, they impact cell behaviour, and their dysregulation is associated with human diseases like cancer, liver fibrosis, nephrotic syndrome, insulin resistance and pathologic neovascularization[16,20,31]. Because inhibiting the GEF activity of GIV and DAPLE blocks their adverse effects when dysregulated in pathologic situations[16,20,31], disruption of their interaction with G proteins is emerging as a promising therapeutic strategy[31].

G proteins have been crystallized bound to representative members of every major class of binding partners: GPCRs; GTPase-activating proteins (GAPs)[32]; guanine-nucleotide dissociation inhibitors[33] and effectors[34,35], but not yet with non-receptor GEFs. In contrast to GPCRs, GBA proteins activate monomeric Gα instead of Gαβγ (refs 18–20,25) and their mechanism of action remains ill-defined. We set out to investigate the molecular basis for the GPCR-independent activation of Gαi3 by proteins of the GBA family, and to establish possible differences and similarities with receptor-mediated activation. By means of nuclear magnetic resonance (NMR) spectroscopy, computational modelling and biochemical assays, we characterize the binding site for GBA proteins and associated structural changes in Gαi3 at the residue level. Our results provide the first detailed view of the molecular mechanism by which proteins with a GBA motif regulate Gαi3, revealing fundamental differences with GPCR-mediated activation of G proteins.

## Results

### GIV binds similarly to GDP-bound and nucleotide-free Gαi3.
We focused our initial efforts on GIV, the first-identified and best-characterized member of the GBA motif-containing family of non-receptor GEFs. GPCRs bind preferentially to nucleotide-free G proteins (Gα-[ ]) over G-GDP and dissociate from G-GTP. We analysed the ability of GIV to bind different states of Gαi3 along the activation pathway, that is, Gαi3-[GDP]→Gαi3-[ ]→Gαi3-[GTP], by depleting purified Gαi3 of nucleotides and reloading it (or not) with GDP or GDP·AlF$_4^-$ (which mimics the GTP-bound transition state; Fig. 1b). Consistent with previous results[18], we found that GIV binds to Gαi3-[GDP] but not to Gαi3-[GTP] (Fig. 1b). Moreover, GIV binds equally to Gαi3-[ ] and Gαi3-[GDP] (Fig. 1b). To rule out that this observation was due to inefficient GDP depletion, we used Ric-8A as a control. The binding preference of Ric-8A for different states of Gαi mimics that of GPCRs[22], that is, Gαi3-[ ] > > > Gαi3-[GDP] > Gαi3-[GTP]. Ric-8A bound much more to Gαi3-[ ] than to Gαi3-[GDP] (Fig. 1b), indicating that nucleotide depletion occurred efficiently. GAIP (also known as RGS19) was used as a control to rule out that the nucleotide depletion procedure compromised Gαi3 integrity. GAIP is a GAP and, as expected[36], it bound to Gαi3-[GTP] but not to Gαi3-[GDP] (Fig. 1b), demonstrating that nucleotide-depleted Gαi3 can bind nucleotides and adopt an active conformation. These results indicate that GIV, contrary to GPCRs, binds to Gαi3-[ ] and Gαi3-[GDP] to a similar extent. An independent demonstration that GIV binds similarly to Gαi3-[GDP] and Gαi3-[ ] was obtained by co-immunoprecipitation experiments in mammalian cells. Full-length GIV co-immunoprecipitated with a G-protein mutant unable to bind nucleotides, thus mimicking Gαi3-[ ][37], as efficiently as with the wild-type (WT) protein (Supplementary Fig. 1). These results indicate that the association of GIV with G proteins along the activation pathway differs from that of GPCRs (Fig. 1c): GIV binds with high affinity to monomeric Gαi-[GDP], remains bound to Gαi with similar affinity on nucleotide release and eventually dissociates on GTP binding to release the active G protein.

### NMR reveals that GIV perturbs discrete regions of Gαi3.
A synthetic 16-mer GEF peptide (named KB-752) has been previously crystallized in complex with Gαi1-GDP[38]. The sequence of this peptide is not present in any human protein but has similarity to the GBA motif (Supplementary Fig. 2). However, comparison of Gαi1-GDP free or bound to this peptide shows that the nucleotide-binding pocket remains essentially unchanged (Supplementary Fig. 2), providing limited insight into the mechanism by which GBA proteins activate Gαi. This might be due to loss of structural rearrangements in Gαi under crystallization conditions. To gain further insights into how a naturally occurring GBA sequence binds to Gαi and structural rearrangements associated with it, we investigated the molecular mechanism of Gαi3 activation by GIV using solution NMR spectroscopy. Perturbation in the NMR signals of the backbone amide groups of Gαi3 protein on ligand binding should inform about direct intermolecular contacts as well as indirect structural rearrangements, providing information at the residue level.

Full-length GIV is 1,870-amino acid long and is composed of multiple domains (Supplementary Fig. 2). Attempts to purify the >220 kDa full-length protein from bacteria gave extremely poor yields of low-quality protein, making its use impossible for NMR experiments. As an experimentally tractable alternative, we used a C-terminal fragment of GIV corresponding

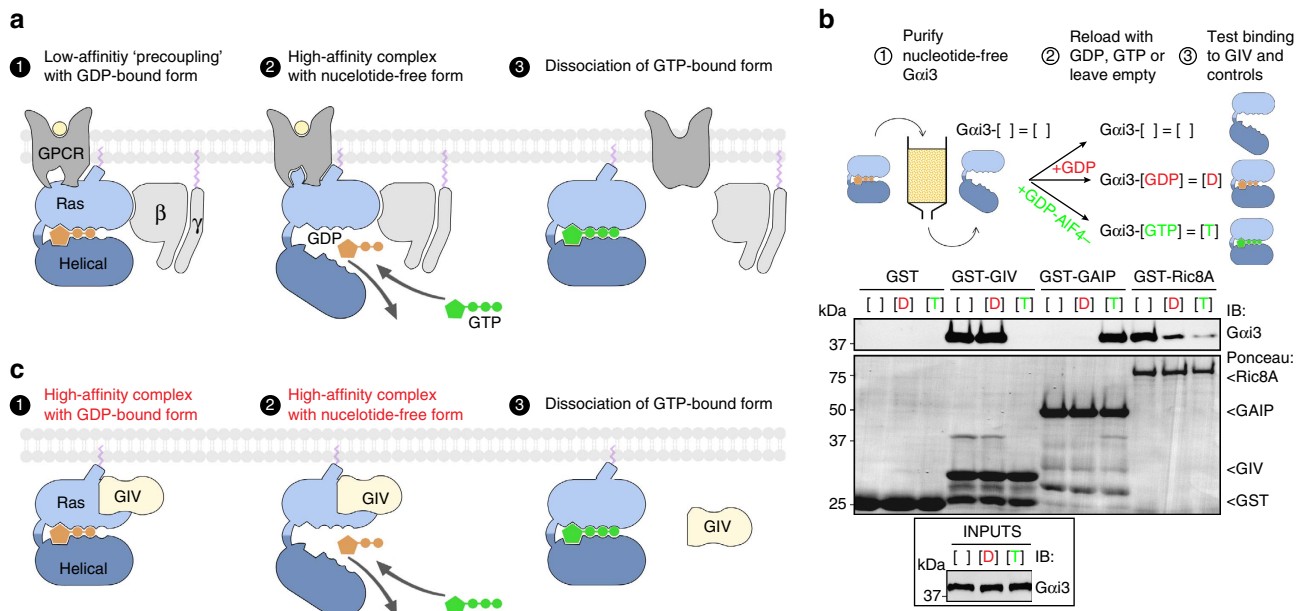

**Figure 1 | GIV binds to GDP-loaded and nucleotide-free Gαi3 to a similar extent but does not bind to GTP-loaded Gαi3.** (**a**) Sequential steps in GPCR-mediated activation of G proteins. GPCRs bind with low affinity to Gαβγ-GDP (1), promote GDP release and form a high-affinity complex with the nucleotide-free G protein (2) and dissociate on GTP loading (3). Dissociation of the GPCR–G protein complex ensures that the active signalling species Gα-GTP and free Gβγ proceed to act on their effectors. The Ras-like and all-helical domains of Gαi3 are shown in light and dark blue, respectively. (**b**) GIV binding to Gαi3 in different conformations of its activation pathway. Top, Gαi3 was nucleotide-depleted and maintained nucleotide-free (Gαi3-[ ]) or reloaded with GDP (Gαi3-[GDP]) or GDP/AlF$_4^-$ (Gαi3-[GTP]). Bottom, pull-down assays showing equal binding of GIV to Gαi3-[ ] and Gαi3-[GDP] but no binding to Gαi3-[GTP]. GAIP binds exclusively to Gαi3-[GTP] and Ric-8A binds preferentially to Gαi3-[ ], followed by weaker binding to Gαi3-[GDP] and dissociation from Gαi3-[GTP][21,22]. One experiment representative of at least three is shown. (**c**) Sequential steps in GIV-mediated activation of G proteins. The substrate for GIV's GEF activity is monomeric Gαi (ref. 18) instead of trimeric Gαβγ as for GPCRs. In contrast to the steps of GPCR-mediated activation, GIV binds to Gα already in its GDP-bound conformation (1) and remains bound with similar affinity to the nucleotide-free conformation after promoting GDP release (2). As in the case of GPCRs, the GEF–G protein complex dissociates on spontaneous loading of GTP on nucleotide-free Gα (3).

to the last 210 amino acids (that is, GIV-CT, residues 1,660–1,870). This fragment was chosen for several reasons: (i) a thorough characterization of GIV-CT has revealed that it recapitulates the biological properties of the full-length protein in mammalian cells, including G-protein-dependent signalling[27,29]; (ii) it contains the GBA motif and mutation of this motif in the context of either full-length GIV or GIV-CT leads to loss of G-protein binding and G-protein-dependent signalling[18,27,29,30]; (iii) GIV-CT is sufficient to bind and activate G proteins *in vitro*[25], and G-protein binding and/or activation is comparable to that obtained with larger fragments (for example, residues 1,425–1,870 translated *in vitro*[18] or 1,623–1,870 purified from bacteria[25]). Thus, GIV-CT is a reliable approximation to the G-protein-binding properties of the native protein and its signalling functions in cells.

NMR spectra of Gαi3-GDP were recorded in the absence or presence of GIV-CT. Backbone amide signals of Gαi3-GDP, as well as the side chain indol signals of its three tryptophans (W131, W211 and W258), were assigned in the two-dimensional TROSY and three-dimensional (3D) HNCO spectra by comparison with the published assignments under the same experimental conditions[39]. As expected from the increase in molecular weight and the binding of a non-deuterated ligand, there was an overall decrease in signal intensity (median ~3-fold) when NMR measurements were carried out in the presence of GIV-CT (residues 1,660–1,870, containing the GBA motif). However, the effect was more pronounced for certain signals, with some even disappearing (for example, W211 and W258; Fig. 2a,b). This may be due to appearance of the signal of the bound form at different frequencies

that we were not able to identify, or to a change in the local dynamics that broadened the signal beyond detection. Regardless of the cause, a large intensity loss reflects perturbations specifically caused by GIV binding and it was quantified for each signal by the calculation of the intensity ratio between the free and bound forms ($I_{ratio}$). In addition, we measured significant chemical shift perturbations on GIV-CT binding for a fraction (<20%) of the assigned peaks (Fig. 2a,b). Assignment of the signals in the bound form was made on a nearest-neighbour basis in both the two-dimensional and 3D spectra, and was confirmed by the similar pattern of signals observed in the presence of the shorter GIV fragment 1,671–1,696 (GIVpept; Supplementary Fig. 3). Assignments of Gαi3-GDP bound to GIVpept were confirmed by a titration with the peptide. The overall decrease in signal intensities on binding of GIVpept was only modest (~15%), which increases the confidence of the identified perturbations, especially for the $I_{ratio}$. The results revealed a high correlation between the perturbations induced by GIV-CT and GIVpept (Supplementary Fig. 3c,d), which indicates that the GBA motif of GIV is causing most of the perturbations observed with GIV-CT. The similar behaviour observed for the short and long fragments of GIV is consistent with the disordered nature of the 210-residue-long C-terminal region of GIV (Supplementary Fig. 4). Circular dichroism shows little, if any, secondary structure, and no cooperative transition is observed in the thermal denaturation, indicating the absence of a tertiary structure. The NMR spectrum confirms that the C-terminal region of GIV is intrinsically disordered as the backbone amide signals show very little dispersion in the proton frequency dimension.

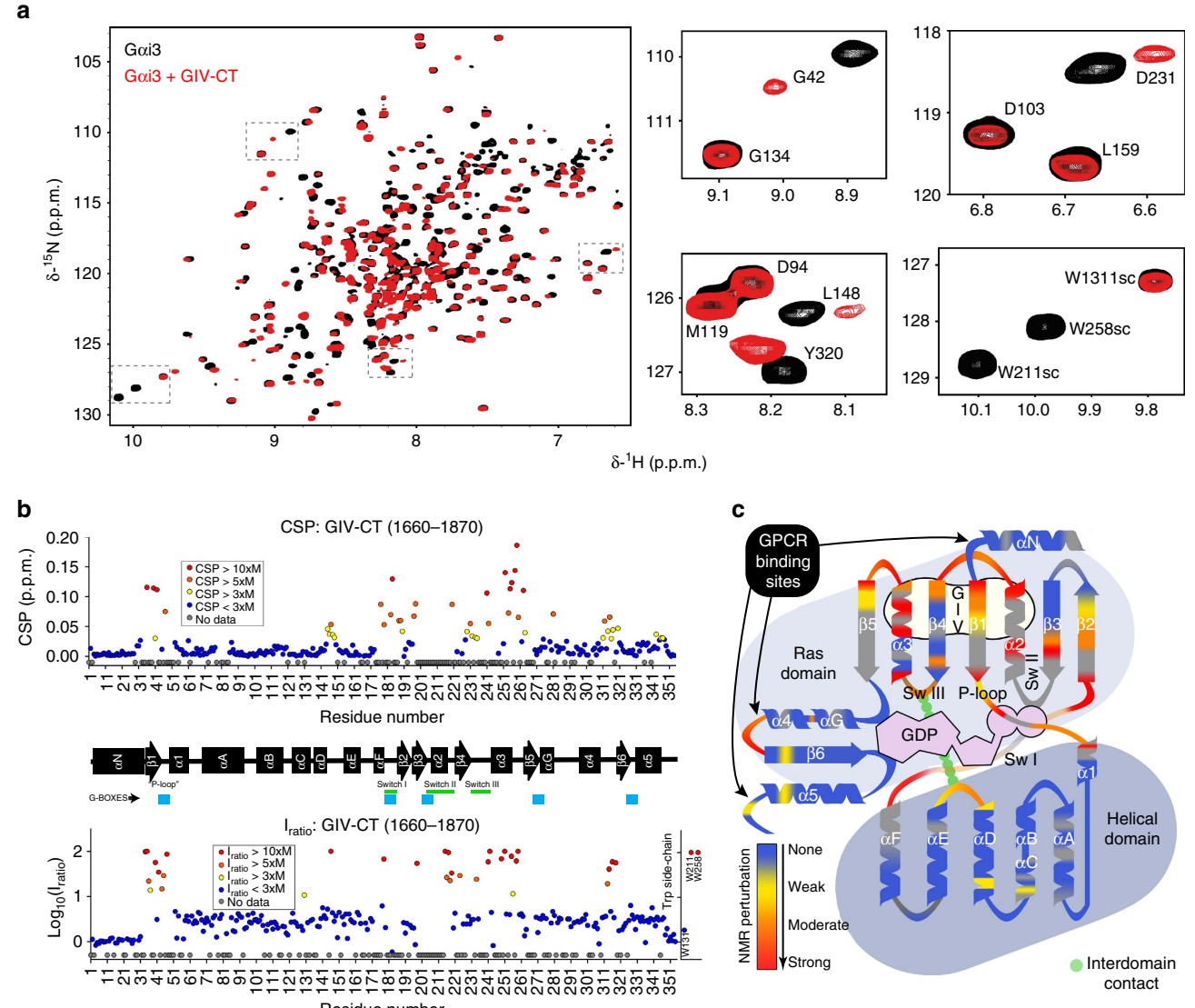

**Figure 2 | NMR mapping of GIV binding on Gαi3.** (**a**) Overlay of $^1$H–$^{15}$N TROSY spectra of Gαi3 free and bound to GIV-CT. Left, $^1$H–$^{15}$N TROSY spectra of $^2$H,$^{13}$C,$^{15}$N-Gαi3 in the absence (black) or presence of His-GIV-CT (1660–1870) (red). Right, selected regions from the overlaid spectra depicting representative perturbations in Gαi3 signals induced by GIV binding. Some signals (top) exhibit chemical shift changes on GIV binding. Some other signals (bottom) exhibit marked signal intensity reductions on GIV binding (see text for interpretation). (**b**) Quantification of GIV-induced perturbations of the NMR signals of Gαi3. Chemical shift (CSP, top graph) or intensity perturbations ($I_{ratio} = I_{bound}/I_{free}$, bottom graph) of the backbone amide signals of the TROSY NMR spectra in **a**. Red, orange and yellow circles indicate residues undergoing perturbations larger than 10, 5 or 3 times the median (M), respectively. Blue circles indicate Gαi3 residues with perturbations smaller than three times the median (M) and grey circles residues for which no reliable NMR measurement could be made. The horizontal black bar in the middle depicts the secondary structure elements of Gαi3 derived from our homology model (see below), and is annotated with the position of the three switch regions (green) that undergo conformational changes on GTP binding and the five conserved G-box sequences (blue) that mediate nucleotide binding. (**c**) Schematic representation of GIV-induced NMR perturbations on Gαi3. Gαi3 secondary structure elements are depicted with a relative orientation to mimic their position with respect to the nucleotide in the three-dimensional structure. Colour coding is the same as in **b** and corresponds to a composite of CSP and $I_{ratio}$ perturbations. For residues in which information for both CSP and $I_{ratio}$ perturbations was available, the largest of the two was taken. GDP is shown in purple and the interaction between the Switch III of the Ras-like domain and the αD–αE loop of the all-helical domain is shown in green. The GIV-binding region predicted by modelling is shown in light beige and the GPCR-binding elements are marked by arrows.

As a first approach to interpret the NMR measurements, which cannot unambiguously differentiate between direct contacts and indirect structural rearrangements, we analysed them in conjunction with molecular modelling. For this, a 3D structural model of the GBA motif of GIV (residues 1,678–1,696) bound to Gαi3 was built based on homology with the KB-752/Gαi1 crystal structure, *ab initio* extension of the GBA sequence and docking (Fig. 3). This model provides independent information because

no constrains based on the NMR data were applied to build it. Although this model does not capture the possible conformational changes distant to the peptide-binding site (absent in the model template), it informs of the protein–protein interface and provides a framework to visualize what specific perturbations might be due to direct GBA binding or indirect conformational changes. The NMR perturbations mapped predominantly to the Ras-like domain (Figs 2c and 3, and Supplementary Fig. 3d).

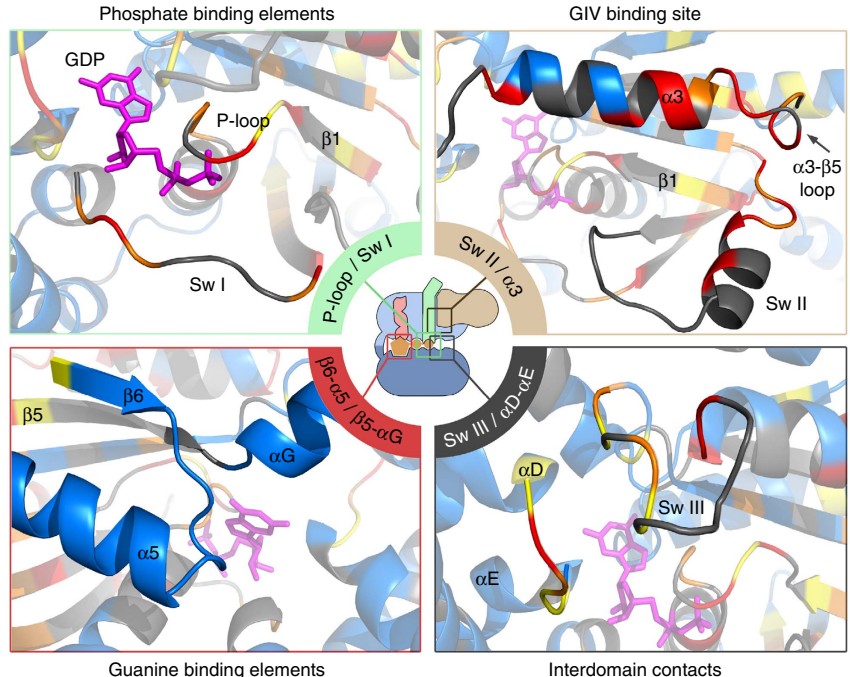

**Figure 3 | Mapping the GIV-induced NMR perturbations onto a structural model of Gαi3 in complex with the GEF motif of GIV.** A model of GIV (residues 1,678–1,696) bound to Gαi3 was generated based on homology with the Gαi1/KB-752 complex[38] and *ab initio* C-terminal extension of GIV. Residues were colour-coded for NMR perturbations as in Fig. 2. A cartoon diagram of Gα is shown in the middle and the detail of four specific regions of the structural model are enlarged: I, P-loop/Switch I, containing the G-1 and G-2 boxes that bind the nucleotide phosphates; II, Switch II/α3, corresponding to the predicted GIV-binding site on Gαi3 (Fig. 4); III, Switch III/αD-αE loop, which stabilize interdomain interactions that favour nucleotide retention; IV, β5–αG and β6/α5 loops, corresponding to the G-4 and G-5 boxes that bind the nucleotide base ring.

We observed extensive perturbations in the GIV-binding site seen in the computational model, which is formed mainly by the α3 helix, α3/β5 loop and the Switch II (SwII) region (Figs 2c and 3). There are additional perturbations in the nucleotide-binding site, which made no direct contact with GIV in our model. More specifically, all the residues of the P-loop and the Switch I (SwI) region for which we could measure NMR data displayed significant perturbations (Figs 2c and 3). The P-loop and SwI contain the G-1 and G-2 boxes, two of the five conserved nucleotide-binding elements of G proteins[3]. The G-1/P-loop and G-2/SwI are responsible for binding the phosphate moieties of GDP, whereas G-4 and G-5 bind the region of the nucleotide base (G-3 makes direct contact with GTP but not with GDP). Interestingly, G-4 (that is, β5–αG loop) and G-5 (β6–α5 loop) remained unperturbed by GIV binding (Figs 2c and 3). The selective effect of GIV on the phosphate-binding elements indicates a divergence from the mechanism of action of GPCRs, which alter all four G-1, G-2, G-4 and G-5 on activation[10,11,13]. We also observed moderate to strong perturbations in the interdomain interface of Gαi3, that is, the αD–αE loop (helical domain) and SwIII (Ras-like domain; Figs 2c and 3). These perturbations are compatible with a local disruption of the interdomain interface, which might be related to the domain separation that accompanies increased nucleotide exchange on the action of both GPCRs[4,5,8,10,14] and the non-receptor GEF Ric-8A (ref. 23).

These results suggest an allosteric mechanism of action for GIV, whose binding to the α3/SwII cleft induces conformational changes in the phosphate-binding region of the nucleotide-binding pocket that weaken GDP binding. The most direct communication route between the GIV-binding site and the nucleotide-binding pocket is the β1 strand, which extends from the bottom of the α3/SwII cleft, where GIV binds, into the P-loop.

Consistent with this hypothesis, the β1 strand undergoes strong NMR signal perturbations on GIV binding (Figs 2c and 3, and Supplementary Fig. 3d). This strand has also been proposed to serve as an allosteric route for GPCR-mediated activation[10].

**Characterization of Gαi3 binding to GIV by mutagenesis.** To further validate the mode of action of GIV on Gαi3, we carried out a biochemical analysis by site-directed mutagenesis. We reasoned that mutation of residues in direct contact with GIV should show a larger impact on binding than mutation of residues involved in indirect structural rearrangements. The design of the mutants was based on the NMR and modelling data, but also on *in silico* thermodynamics analysis of the Gαi3–GIV model, which predicts the Gαi3 residues that contribute the most to the complex formation (Fig. 4a,b). A detailed rationale of the mutant design is described in Supplementary Note 1. For the characterization of these mutants, we used purified His-tagged Gαi3. The tagged protein binds the GBA motif of GIV with the same affinity as the untagged Gαi3 used in the NMR experiments and is activated to similar levels (Supplementary Fig. 5). All mutants were purified with similar yields and purity (Supplementary Fig. 6a), and were capable of adopting an active conformation on GTPγS binding (Supplementary Fig. 6b), indicating that none of them had major structural defects.

We found reduced GIV binding for mutants of any of the Gαi3 residues in SwII, α3 or the α3/β5 loop that were predicted by the thermodynamics analysis to contribute to GIV binding and also displaying NMR signal perturbations on GIV binding. These residues are W211, F215, L249, W258 and F259 (Fig. 4c,d, and Supplementary Figs 6 and 7). On the other hand, mutation of V218 or K257, two residues in the same region but predicted to marginally destabilize GIV binding, had little effect on binding.

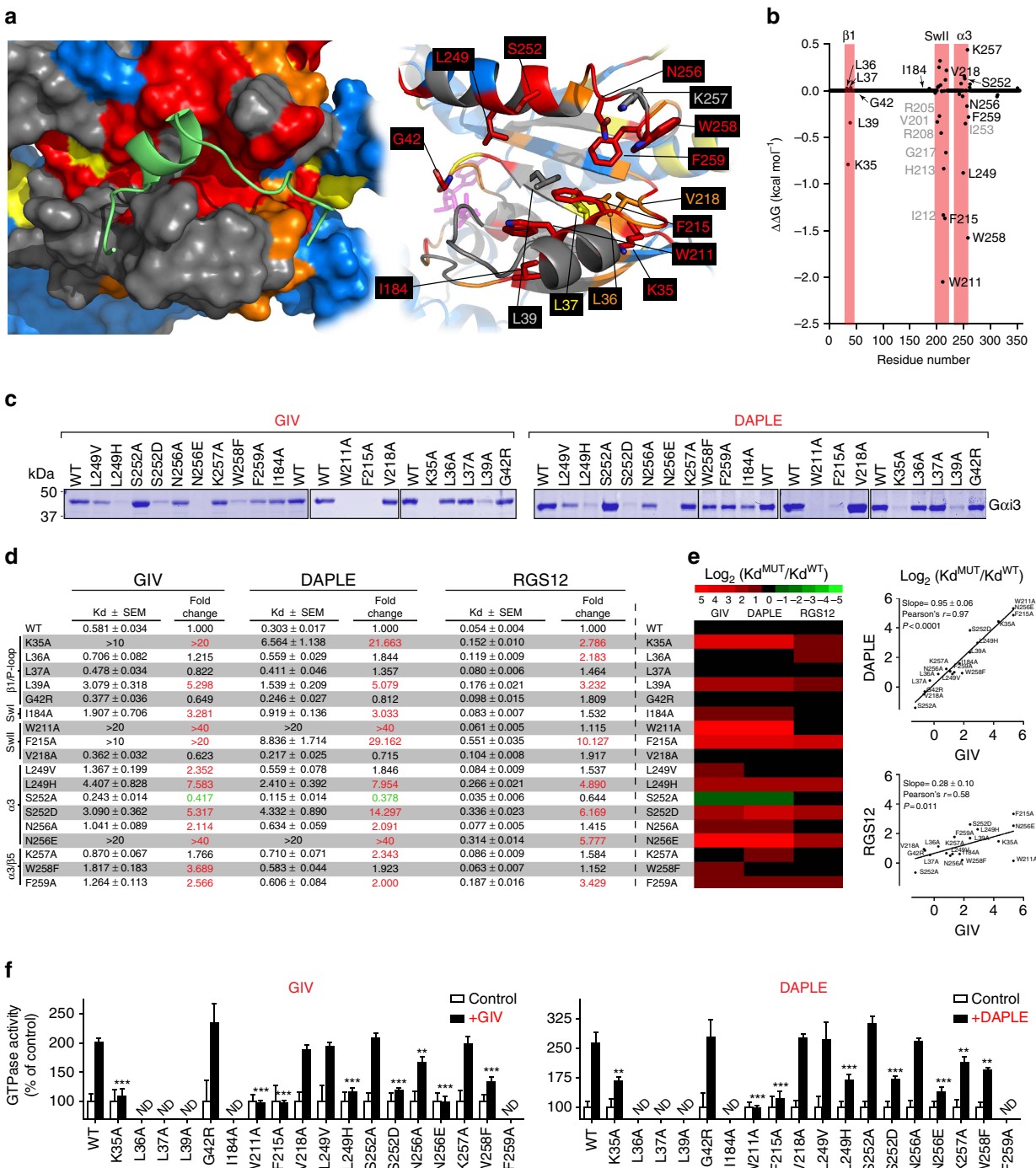

**Figure 4 | Characterization of the GIV-binding site on Gαi3.** (**a**) GIV-binding site on Gαi3 and location of residues selected for mutagenesis. Left, representation of the molecular model of Gαi3 bound to GIVpept (green ribbon) coloured as in Fig. 3. Right, ribbon representation of the view of Gαi3 on the left with residues selected for mutagenesis shown in sticks. (**b**) Calculation of energetic contribution of individual Gαi3 residues to the interaction with GIV. The Gαi3/GIV model was analysed using FoldX. Experimentally mutated residues are indicated with black labels. Residues predicted to stabilize the interaction but not experimentally tested by mutagenesis are labelled grey. (**c**) GIV and DAPLE binding to Gαi3 mutants in pull-down assays. Binding of His-Gαi3 WT or mutants to immobilized GST-GIV (left) or GST-DAPLE (right) was visualized by Coomassie blue staining. One experiment representative of at least three is shown. GST protein loading and negative controls are shown in Supplementary Fig. 5. (**d**) Equilibrium dissociation constants ($K_d$) for the binding of Gαi3 mutants to the GBA motif of GIV and DAPLE or the GoLoco motif of RGS12. $K_d$ values for fluorescein-labelled peptides derived from GIV, DAPLE or RGS12 were calculated from fluorescence polarization measurements (complete binding curve data sets in Supplementary Fig. 6). Mean ± s.e.m., $n = 3–7$. Fold changes in the $K_d$ larger than 2 or smaller than 0.5 are indicated in red or green, respectively. (**e**) The effect of Gαi3 mutations on GIV binding is highly correlated with DAPLE binding but not with RGS12 binding. Left, heat map (green to red scale shown on top) of $Log_2$ $K_d$ ratios for each Gαi3 mutant relative to WT. Right, correlation plots of $Log_2(K_d^{MUT}/K_d^{WT})$ values for GIV and DAPLE (upper right) or GIV and RGS12 (lower right). (**f**) GIV- and DAPLE-mediated activation of Gαi3 mutants. Steady-state GTPase activity of His-Gαi3 WT or mutants in the absence (white) or presence (black) of GIV (left) or DAPLE (right). Mean ± s.e.m., $n ≥ 3$, **30–50% inhibition, *** > 50% inhibition of GIV/DAPLE-mediated activation compared to Gαi3 WT. Raw basal activities in the absence of GIV or DAPLE are shown in Supplementary Fig. 7. ND, not determined due to severely compromised basal activity.

Mutation of S252 and N256, two residues in the α3 helix displaying large NMR perturbations (Fig. 2) but not predicted to stabilize GIV binding, severely impaired GIV binding when mutated to D and E, respectively, but not when mutated to A (Fig. 4c,d, and Supplementary Figs 6 and 7). These two residues are located in the GIV-binding site, and mutation to D or E, respectively, creates a steric/electrostatic hindrance for GIV binding that does not occur when mutated to A. Mutation of I184 in SwI also diminished binding (Fig. 4c,d). This residue is not predicted to contribute directly to GIV binding (Fig. 4a,b) but makes a direct intramolecular contact with SwII based on our homology model (Fig. 4a). On the other hand, I184 undergoes strong NMR signal perturbations in the presence of GIV (Fig. 2 and Supplementary Fig. 3). This is consistent with I184 interacting with SwII on GIV binding and possibly stabilizing a conformation of SwII that favours GIV binding. These mutagenesis results indicate that the groove formed by α3, SwII and α3/β5 loop elements is the direct binding site for GIV.

To further characterize the Gαi3/GIV-binding site, we analysed a battery of GIV mutants for binding to Gαi3 in a peptide array format (Supplementary Fig. 8). GBA motif peptides in which nine selected positions were replaced by every other natural amino acid one at a time were probed for Gαi3 binding (Supplementary Fig. 8b). Selected mutants were tested for confirmation of binding in solution (Supplementary Fig. 8C). The results revealed that mutation of GIV residues predicted by our model to be located in the vicinity of Gαi3 residues important for binding (as determined by mutagenesis) efficiently disrupted the binding. On the other hand, mutation of residues in the same region of GIV but not predicted to be important for Gαi3, had marginal or no effect on binding (see Supplementary Note 2 for a full description). These results further validate our model of GIV binding to a pocket on Gαi3 formed by the α3, SwII and α3/β5 loop.

Mutation of Gαi3 residues in the β1/P-loop axis predicted by the thermodynamics analysis in Fig. 4b to contribute to GIV binding (K35 and L39) also impaired GIV binding in pull-down or fluorescence polarization assays (Fig. 4c,d, and Supplementary Figs 6 and 7). On the other hand, mutation of residues in the same region and displaying NMR perturbations, but not making direct contact with GIV (L36, L37 and G42) did not significantly affect binding (Fig. 4c,d). Because G42 is located in the middle of the P-loop, this suggests that GIV induces indirect structural rearrangements in the nucleotide-binding pocket.

We monitored GIV-mediated enhancement of Gαi3 steady-state GTPase activity (which reflects its GEF activity[25]) for the mutants described above. Not all mutants could be analysed due to compromised intrinsic (basal) activity (Supplementary Fig. 7b), but the results with the remaining mutants were in excellent agreement with their relative ability to bind GIV (Fig. 4f). The activity of selected mutants relative to WT was confirmed by GTPγS-binding assays, which measure nucleotide exchange more directly (Supplementary Fig. 7b,c).

Thus, the mutagenesis analysis is highly congruent with the mode of action suggested by the NMR and molecular modelling data. Together, these mutagenesis results indicate that GIV binding and subsequent G-protein activation require residues located in a pocket of the Ras-like domain where NMR signal perturbations are clustered (that is, SwII, α3 and α3/β5 loop). On the other hand, the observation of additional NMR signal perturbations in residues of the β1/P-loop that are not crucial for GIV binding point to indirect structural rearrangements associated with its ability to modulate nucleotide exchange.

**Different GBA proteins bind similarly to Gαi3.** DAPLE has been recently shown to bind and activate Gαi3 via a GBA motif

similar to that in GIV[20]. We reasoned that if the G-protein-binding mode of different GBA motifs is conserved, the effect of specific Gαi3 mutations on DAPLE–Gαi3 coupling should closely recapitulate the observations with GIV–Gαi3. Indeed, we found that the effects of Gαi3 mutations in pull-down (Fig. 4c), fluorescence polarization (Fig. 4d,e) and G-protein activity assays (Fig. 4f and Supplementary Fig. 7b,c) were highly correlated with those of GIV. To assess specificity, we tested the same set of mutants for binding to a peptide derived from RGS12's GoLoco motif. We chose RGS12 GoLoco motif as a stringent test for specificity because it also docks onto the α3/SwII cleft of Gαi-GDP[33]. Consistent with binding to this common site, we found that some of the mutations that impair GIV binding also impair binding of the RGS12 GoLoco motif (Fig. 4d,e). However, the overlap of the sets of Gα mutations that affect binding to GBA or GoLoco motifs is only partial, and many of the mutations that affect binding to both motifs do so to a significantly different extent. The divergence of the effect of same Gα mutants on GoLoco or GBA binding is in agreement with the comparison of the crystal structures of Gαi bound to a GoLoco motif or the GBA-like peptide KB-752, which showed that the common binding site formed by the α3 helix and SwII region adopts a different conformation[38]. The correlation of the effect of the entire set of Gαi3 mutants on the dissociation constant of RGS12–Gαi3 versus GIV–Gαi3 is markedly reduced (Pearson's = 0.58) compared to the correlation with DAPLE–Gαi3 (Pearson's = 0.97; Fig. 4d,e). A notable example is mutant W211A, which completely abolishes GIV and DAPLE binding but has no effect on RGS12 binding. Thus, the overall effect of the mutants indicates that different GBA proteins share a common and specific site of binding to G proteins, which partially overlaps with the GoLoco-binding site but engages Gαi3 in a different way.

**GBA proteins and GPCRs couple differently to G proteins.** Figure 5 depicts the mapping of the NMR perturbations induced by GIV on Gαi3 and the GPCR contact sites reported for a rhodopsin–G-protein model[12]. Previous studies using crystallographic, biophysical, biochemical and computational approaches indicate that GPCRs make contact with the C-terminal tail of Gα subunits as well as with other regions such as the boundary between the N-terminal helix (αN) and the β1 strand, and the α4/β6 loop[9–12] (Fig. 5b). These molecular contacts have been proposed to transmit allosteric conformational changes to the nucleotide-binding pocket via two distinct routes[10,12–14]. One route goes through the α5 helix (which connects the C-terminal tail to the β6/α5 loop that binds the nucleotide guanine ring) and the other one through the β1 strand (which connects αN to the P-loop that binds the nucleotide phosphates; Fig. 5c). Our results suggest that while GIV induces structural changes along the β1/P-loop axis (Fig. 5c), it does not perturb significantly the C terminus, α5 or guanine-binding loops (Fig. 5a).

A subset of residues with GIV-induced NMR perturbations mapped to the α4/β6 loop (Fig. 6a), one of the GPCR-binding sites, but mutation of two of those residues (R313 and K317) did not affect Gαi3 binding and activation by either GIV or DAPLE (Fig. 6b–d). These observations indicate the NMR signal perturbations observed in the α4/β6 loop are likely due to a secondary structural rearrangement, perhaps mediated by the α3/β5 loop that directly binds GIV (Fig. 6a). We also tested a mutant lacking the last nine residues of Gαi3 (ΔC9), which removes the main GPCR-binding site, and observed no effect on binding and activation by GIV or DAPLE (Fig. 6b–d).

The results described above indicate that Gαi3 mutants that interfere with GPCR coupling have no effect on how GBA

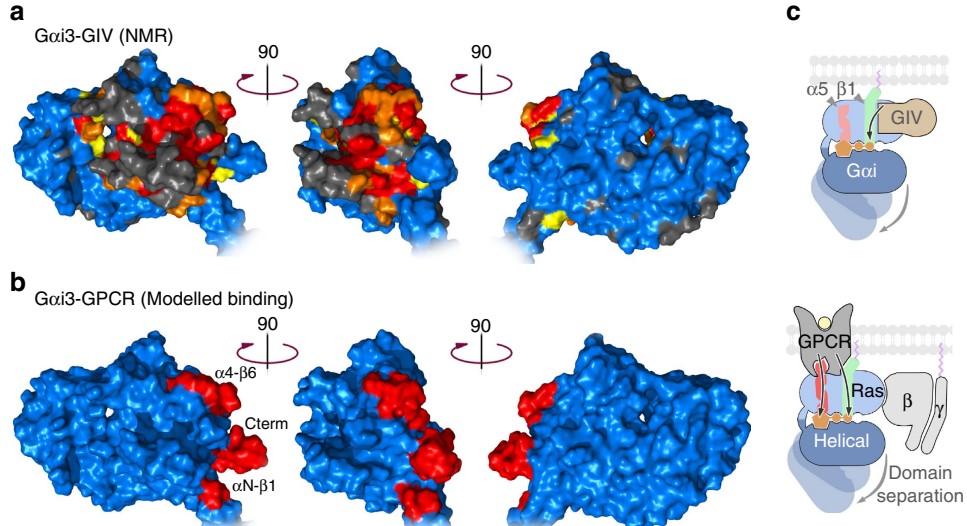

**Figure 5 | Comparison of GIV and GPCR binding to Gαi3.** (**a**) Mapping of GIV-induced NMR perturbations onto the structure of Gαi3. The overlay of NMR perturbations on a structural model of GIV-bound Gαi3 was generated as described in Fig. 3. (**b**) Mapping of GPCR contact sites on the structure of Gαi3. Gαi regions contributing to the interaction with GPCRs were extracted from a previously described model of Rhodopsin-Gαi1 (ref. 12) and coloured red on the Gαi3 structure model described in **a** (in blue). (**c**) Proposed allosteric routes for GIV- and GPCR-mediated acceleration of nucleotide exchange on Gα. Top, GIV binds to a cleft delimited by the SwII and α3 helix of Gα and induces perturbations in the β1 strand located at the bottom of this cleft that are allosterically transmitted to the contiguous P-loop in the nucleotide-binding pocket. Bottom, GPCRs engage the C-terminal α5 helix, the α4/β6 loop and the αN–β1 linker of Gα to transmit conformational changes to the nucleotide-binding pocket via two major routes: (i) to the guanine-binding β6–α5 loop via the C-terminal α5 helix; and (ii) to the phosphate-binding P-loop via the β1 strand.

proteins couple to Gαi3. Next we tested whether the converse is also true, that is, mutants that disrupt GIV and DAPLE coupling to Gαi3 have no effect on GPCR-mediated G-protein activation. We used purified bovine rhodopsin and retinal Gβγ (Gβ$_1$γ$_1$) in combination with Gαi3 mutants (Fig. 7a), a previously validated system to monitor GPCR-mediated activation of G proteins in vitro[5–7,11,12,40]. We selected the Gαi3 mutants that most markedly impair GIV and DAPLE binding (K35A, W211A, F215A, L249H, S252D and N256E), as well as Gαi3 ΔC9, a mutant expected not to be activated by rhodopsin. Because GPCR-mediated activation requires an intact G-protein heterotrimer, we first tested whether any of the Gαi3 mutants had altered Gβγ binding. Both W211A and F215A abolished Gβγ binding (Fig. 7b) and were excluded from subsequent experiments. The other mutants that disrupt GIV and DAPLE binding (K35A, L249H, S252D and N256E) were activated efficiently by rhodopsin, whereas the ΔC9 mutation completely abolished it (Fig. 7c). These in vitro results show that GBA proteins and GPCRs use different molecular mechanisms to couple to G proteins. We also investigated the effect of different G proteins mutants in a genetically engineered yeast strain that lacks endogenous GPCRs and expresses a single Gα subunit (human Gαi3 with the first 35 residues (αN) replaced by the corresponding residues of yeast Gpa1 (Fig. 7d). In this system, the natural pheromone response pathway that leads to activation of the ERK-like kinase Fus3 can only be activated by exogenous GEFs (Fig. 7d). Saccharomyces cerevisiae does not express GoLoco proteins. We found that GIV and DAPLE activated Gαi3 WT and Gαi3 ΔC9 equally, while activation of Gαi3 N256E (NE, see rationale for the choice of this mutant in Methods) was blunted (Fig. 7e). Conversely, the human GPCR A2bR equally activated Gαi3 WT and Gαi3 N256E while activation of Gαi3 ΔC9 was completely blunted (Fig. 7f). These results further confirm the divergence between the mechanisms of GPCR and GBA motif coupling to G proteins.

## Discussion

Here we provide the first detailed characterization of the molecular basis for the activation of trimeric G proteins by non-receptor GEFs containing a GBA motif. Our conclusions are supported by consistent results from three independent and complementary approaches: biophysical (NMR); computational (modelling); and biochemical (mutagenesis). The major conclusion is that the mechanism of G-protein activation by GBA motif-containing GEFs is fundamentally different from the mechanism used by GPCR GEFs. Although results from such NMR experiments cannot be directly compared to the available crystal structure of nucleotide-free G protein in complex with a GPCR[4], they can be interpreted with confidence in light of a wealth of additional information on the mechanism of GPCR-mediated activation obtained from experiments using biophysics[5–9]; biochemistry[10,11]; and computational modelling[12–14]. The current model for GPCRs is that activation is caused by the simultaneous perturbation of structural elements that bind both the nucleotide base (β6–α5 and β5–αG loops) and the nucleotide phosphates (P-loop and SwI). According to this model, GPCRs bind to the C terminus of Gα triggering a conformational change through the α5 helix to perturb the nucleotide base-binding region[4,6,10–13]. Perturbation of the phosphate-binding elements is currently explained by two non-exclusive models: one is that it is a consequence of the rearrangements of the nucleotide base-binding site[4,6,12] while the other holds that it is caused by GPCR binding to the N terminus of Gα and subsequent conformational rearrangements transmitted to the P-loop through the β1 strand[10,12]. Our results indicate that in contrast to the GPCR mechanism, GIV does not induce perturbations in the nucleotide base-binding β6–α5 and β5–αG loops, but does cause structural changes in the P-loop and SwI phosphate-binding elements (Figs 2 and 3). Moreover, GIV, as well as other GBA proteins, binds to a Gα region different from the GPCR-binding site. The GBA motif docks in a cavity delimited by the α3/SwII/α3–β5 loop in which the β1 strand lies at the bottom

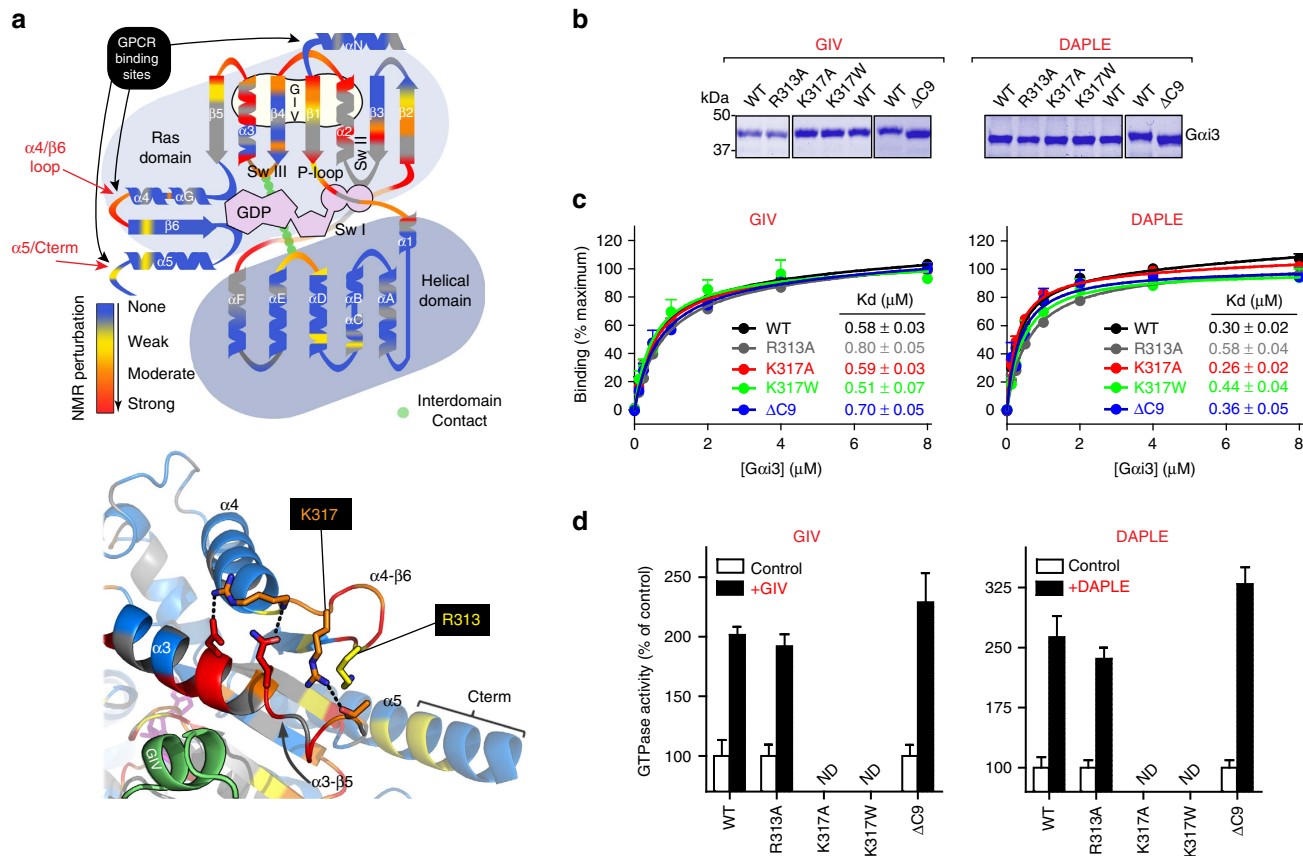

**Figure 6 | The α4-β6 loop and C terminus of Gαi3 are not required for coupling to GIV or DAPLE.** (**a**) GIV-induced perturbations in the α4/β6 loop of Gαi3. Top, diagram of Gαi3 secondary structure elements overlaid with GIV-induced NMR perturbations reproduced from Fig. 2. Bottom, illustration of the position of α4/β6 loop and C-terminal α5 helix relative to the GIV-binding region in the GIV/Gαi3 model. Colour-coded as in the top panel. Two residues selected for mutation are labelled (R313 and K317). Some residues are displayed in sticks to indicate predicted contacts (dotted lines) between the α4/β6 loop and the α3 helix/α3-β5 loop region. The distance between the centroid of the GIV-binding site and the α4-β6 loop and C-distal part of α5 is ∼16 and ∼26 Å, respectively. (**b**) GIV or DAPLE binding to Gαi3 is not affected by mutations in the α4/β6 loop or deletion of the C terminus in pull-down assays. Binding of His-Gαi3 WT or mutants to immobilized GST-GIV (left) or GST-DAPLE (right) was visualized by Coomassie blue staining. One experiment representative of at least three is shown. GST protein loading and negative controls are shown in Supplementary Fig. 5. (**c**) Binding affinity of the GEF motif of GIV or DAPLE for Gαi3 is not altered by mutations in the α4–β6 loop or deletion of the C terminus. Binding of fluorescein-labelled GIV (left) or DAPLE (right) peptides to His-Gαi3 WT or mutants was determined by fluorescence polarization as in Fig. 4d. Curves were fitted to a one site binding model to calculate the $K_d$. Mean ± s.e.m., $n \geq 3$. (**d**) GIV- or DAPLE-mediated activation of Gαi3 is not affected by mutation of the α4/β6 loop or deletion of C terminus. Steady-state GTPase activity of His-Gαi3 WT or the indicated mutants in the absence (white) or presence (black) of GIV (left) or DAPLE (right). Mean ± s.e.m., $n \geq 3$. Raw basal activities in the absence of GIV or DAPLE are shown in Supplementary Fig. 7. ND, not determined due to severely compromised basal activity.

(Fig. 4). The β1 strand directly extends into the P-loop and acts as an allosteric route to perturb the P-loop on GPCR binding[10,12]. We propose that the β1 strand also serves as a conduit to transmit information from the GIV-binding site to the nucleotide-binding pocket. GIV altering only a subset of the nucleotide-binding elements (P-loop/SwI) may be the reason behind the apparent lower GEF efficiency of GIV compared to GPCRs[18]. Nevertheless, this also suggests that weakening phosphate binding is sufficient for GEF-catalysed nucleotide exchange. A recent study[14] indicates that the main driving force for nucleotide release is the structural rearrangement of the Ras-like domain instead of the separation between the Ras-like and all-helical domains as previously suggested[4,5,8,10]. The same study also provided evidence that without destabilization of phosphate binding, nucleotides are not released. However, because GPCRs induce structural rearrangements in all the different nucleotide-binding elements of Gα, it remained unclear whether weakening of phosphate binding was necessary or sufficient to accelerate nucleotide exchange. Our results with the non-receptor GEF GIV highlight the importance of

phosphate binding for nucleotide exchange and suggest that altering the interaction between GDP phosphates and the P-loop/SwI is sufficient to promote nucleotide exchange. This mechanism of G-protein activation regulates signalling in cells, as demonstrated by GIV-dependent signalling via Gαi-GTP[26–28] and free Gβγ[18,20,27,29]. Moreover, GIV and GPCRs activate G-protein signalling in cells to a similar extent as determined by bioluminescence resonance energy transfer-based biosensors[30]. Thus, the moderate GEF activity of GIV in vitro might be either an underestimation of its GEF activity in cells or just sufficient to elicit biological responses. This is in keeping with early observations for the α2 adrenergic receptor, a bona fide Gi GEF, which showed moderate GEF activity in vitro (three to sixfold)[41,42].

The GBA motif of GIV is present in a long disordered region of the protein (Supplementary Fig. 4), and the shorter GBA peptide recapitulated the NMR perturbations caused by GIV-CT (Supplementary Fig. 3), indicating that the GBA motif is sufficient to regulate G proteins. The disordered nature of GIV-CT is intrinsic and not a consequence of the truncation of the protein at

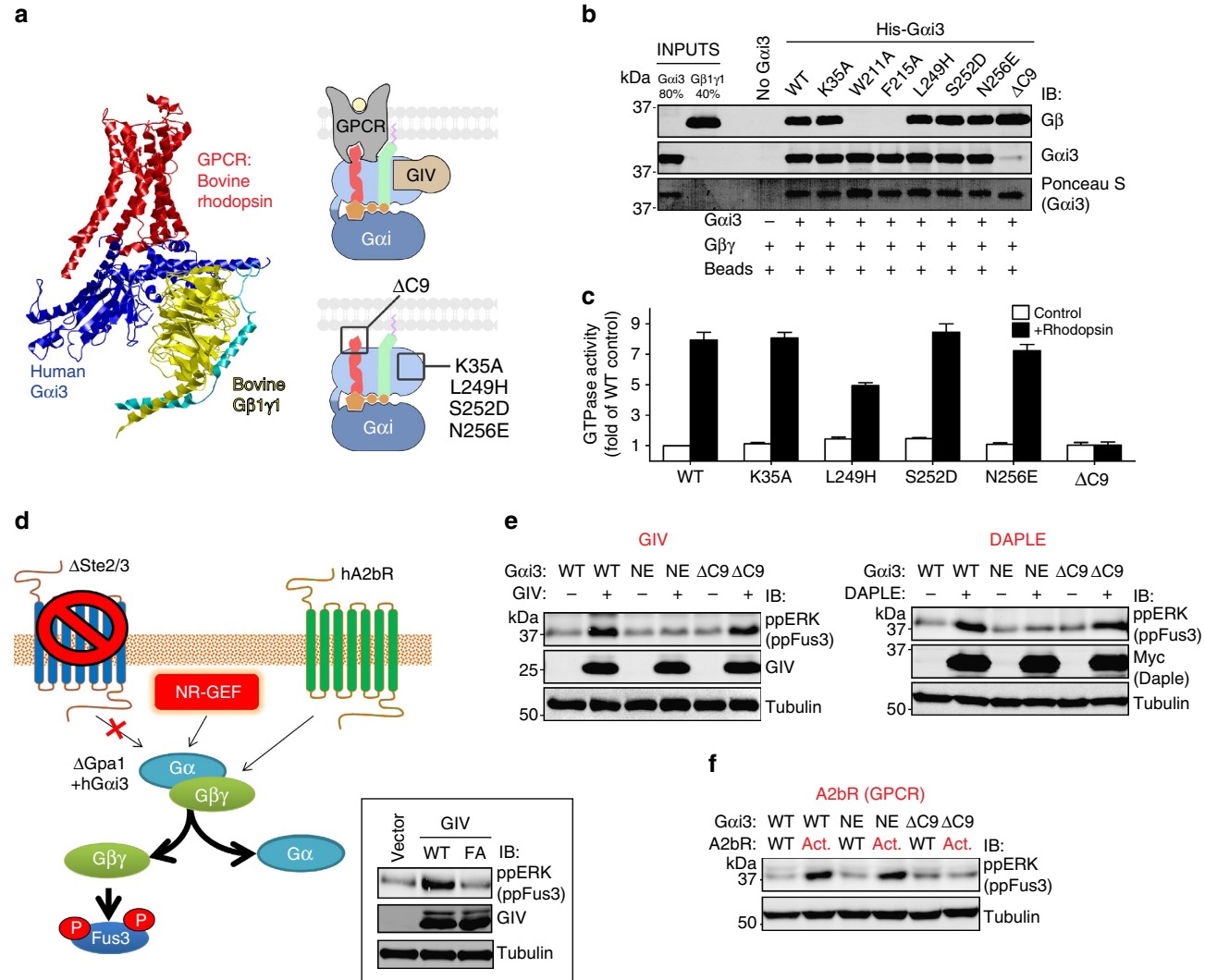

**Figure 7 | Gαi3 mutants that disrupt activation by GIV are activated by GPCRs.** (**a**) System to monitor GPCR-mediated activation of Gαi3 *in vitro*. Left, human His-Gαi3 (blue) was reconstituted with bovine Gβ1γ1 (β: yellow; γ: turquoise) and activated with light-stimulated bovine rhodopsin (red). Diagrams on the right depict the position of mutants relative to the GPCR- and GIV-biding sites. (**b**) Gβγ binding to Gαi3 mutants. His-Gαi3 WT or mutants were incubated with bovine Gβ₁γ₁ and binding determined in pull-down assays. Equal protein loading of Gαi3 ΔC9 (not detected by the Gαi3 antibody) was confirmed by Ponceau S staining (bottom row). One experiment representative of three is shown. (**c**) Gαi3 mutants that disrupt activation by GIV are activated by rhodopsin. Steady-state GTPase activity of His-Gαi3 WT or mutants in complex with Gβ1γ1 was determined in the absence (white) or presence (black) of rhodopsin. Mean ± s.e.m., n = 3. (**d**) Schematic diagram and validation of a yeast-based assay used to monitor G-protein signalling in cells. A genetically engineered *S. cerevisiae* strain was used to determine the levels of G-protein activation on expression of exogenous GEFs like GIV and DAPLE or a GPCR (human A2b receptor) by measuring Fus3 phosphorylation (phospho(pp)ERK antibodies recognize yeast ppFus3). Insert, validation experiment showing activation of human Gαi3 (as determined by Fus3 phosphorylation) in yeast strains expressing GIV WT but not the GEF-deficient F1685A mutant (FA). One experiment representative of three is shown. (**e**) Activation of G-protein signalling by GIV or DAPLE is blocked by the N256E mutation but not by deletion of the C terminus of Gαi3. Yeast strains expressing Gαi3 WT or the indicated mutants in the absence ( − ) or presence ( + ) of GIV (left) or DAPLE (right) co-expression were lysed and immunoblotted as indicated. NE, N256E. One experiment representative of three is shown. (**f**) Activation of G-protein signalling by A2bR is blocked by deletion of the C terminus of Gαi3 but not by the N256E mutation. Yeast strains expressing Gαi3 WT or the indicated mutants and co-expressing A2bR WT or a constitutively active mutant ('Act.', in red) were lysed and immunoblotted as indicated. NE, N256E mutant. One experiment representative of three is shown.

residue 1,660 since disorder is predicted for the C-terminal 330 residues of the GIV sequence, and larger fragments of GIV can also recapitulate G-protein binding and/or activation in a GBA motif-dependent manner[18,25]. The presence in intrinsically disordered regions might be a property common to all GBA motifs identified to date. For example, the GBA motif of both GBAS-1 and DAPLE are located in intrinsically disordered regions[24,43]. In CALNUC, the GBA motif overlaps with the region corresponding to its calcium-binding EF hands. However, Gαi3 binds to the GBA motif

of CALNUC only when the protein is not calcium-bound[19], which is known to result in high disorder of the region comprising the EF hands (and the GBA motif)[44]. Thus, it is possible that the presence of GBA motifs in intrinsically disordered regions facilitates their function by making them more readily accessible to the target G protein.

The molecular mechanism of activation of Gα by GBA motif-containing proteins has more similarities with GEF-mediated activation of small G proteins of the Ras family

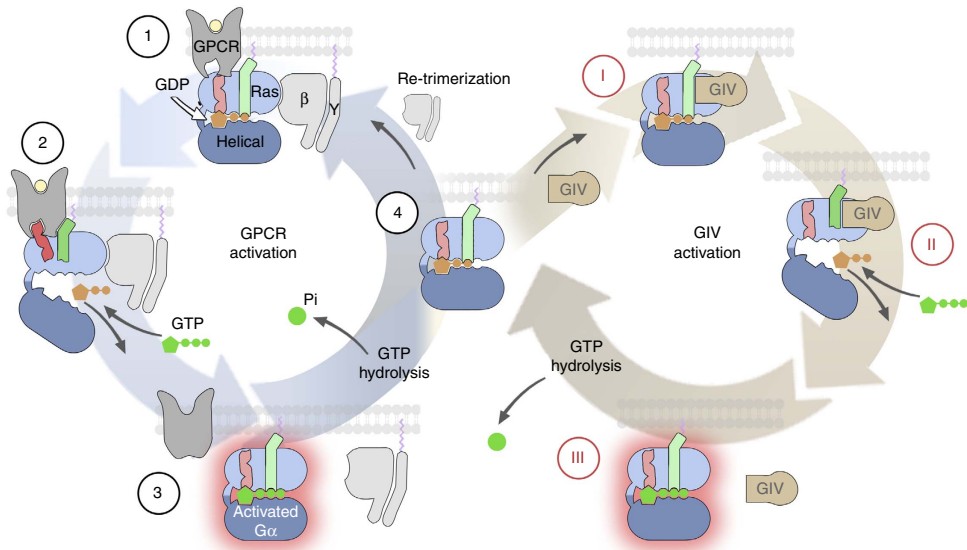

**Figure 8 | Integrated model for the activation of G proteins by GPCRs and non-receptor GEFs of the GBA family.** Left: (1) ligand-bound GPCRs bind to GDP-bound Gα in complex with Gβγ; (2) GPCR binding induces conformational changes in the C-terminal α5 helix (red) and β1 strand (green) of Gα that are transmitted to the nucleotide-binding pocket and facilitate GDP release; (3) after spontaneous loading of GTP, Gα becomes active and dissociates from Gβγ and GPCRs; and (4) Gα hydrolyses GTP into GDP. Gα-GDP re-associates with Gβγ and becomes available for a new cycle of GPCR-mediated activation or binds to GIV (or related non-receptor GEFs) for an alternative cycle of activation (see cycle on the right). Right: (I) GIV binds to monomeric Gα in the GDP-bound state; (II) GIV binding induces conformational changes in the β1 strand (green) of Gα that are transmitted to the P-loop in the nucleotide-binding pocket and facilitate GDP release by loosening binding to the phosphate groups; and (III) after spontaneous loading of GTP, Gα becomes active and dissociates from GIV. When Gα hydrolyses GTP into GDP, Gα-GDP can undergo a new cycle of activation by GIV or re-associate with Gβγ and become available for a new cycle of GPCR-mediated activation (see cycle on the left).

than GPCR-mediated activation of trimeric G proteins. First, GBA proteins displace Gβγ from the trimeric complex[18,20] and exert their GEF activity directly on monomeric Gα (refs 18–20,25), whereas GPCRs work most efficiently as GEFs on intact Gαβγ trimers[40]. Second, GEFs for small G proteins perturb the phosphate-binding region of the G protein while leaving the guanine base-binding region mostly unchanged[45]. Third, as in the case of GEFs for small G proteins, GBA proteins bind to a region that overlaps with the site where effectors and other regulators (like GAPs) bind[13], whereas GPCRs bind to a region that does not overlap.

GBA proteins also differ from the non-receptor GEF Ric-8A. Although both Ric-8A and GBA proteins activate monomeric Gα instead of Gαβγ trimers[21], removal of the C terminus of Gα only abrogates Ric-8A action[22]. This indicates a different mode of G-protein binding for Ric-8A that resembles how GPCR bind. Also, Ric-8A binds preferentially to nucleotide-free Gαi (ref. 22), which again differs from GBA proteins and resembles GPCRs.

The preference of GPCRs for nucleotide-free G proteins[4,21] is considered an event that favours the progression of the reaction in the direction of activation. In the case of GIV, the directionality of the reaction is still ensured by the dissociation from Gαi on GTP binding, but it is unclear why a GEF would bind to Gαi-[GDP] with an affinity comparable to nucleotide-free Gαi. We speculate that such Gαi-[GDP] binding would increase the efficiency of this class of GEFs. To activate Gα, GBA proteins must either displace Gβγ from the trimeric complex or engage transiently formed Gαi-[GDP] *en route* to re-association with Gβγ after a priming cycle of activation. In both scenarios, binding with higher affinity to Gαi-[GDP] would be advantageous to facilitate GBA-mediated activation. Therefore, we propose a model to integrate the GBA-mediated mechanism of activation with the classic G-protein cycle triggered by GPCRs (Fig. 8).

*In vitro* experiments demonstrate that GIV and DAPLE can displace Gβγ (refs 18,20), and GIV can also induce the rapid dissociation of the Gαβγ complex in cells[30].

A question that cannot be unambiguously answered at this point is whether the perturbations observed in the nucleotide-binding pocket in our NMR experiments are due to lower GDP occupancy or structural rearrangements of the GDP-bound form on GIV binding. This is in part because studies in Gαt (ref. 46) and Gαi1 (ref. 47; which shares >90% sequence identity with Gαi3) indicate that the NMR spectra of nucleotide-free Gα do not differ markedly from those of Gα-GDP. Nevertheless, we favour the interpretation that our NMR experiments reflect changes in GDP-bound G proteins because they were carried out in the presence of 300 μM GDP, a 3- or 10-fold molar excess over Gαi3 (for GIV-CT or GIVpept, respectively). Since G proteins bind nucleotides with submicro-molar–nanomolar affinity[48,49], we think that it is unlikely that the GEF activity of GIV will be sufficient to lower it over three orders of magnitude to result in a marked decrease of nucleotide occupancy. Moreover, an overall decrease in the nucleotide occupancy would also be expected to result in perturbations in the guanine-binding elements of the nucleotide-binding pocket, which we do not observe. Thus, our results probably reflect the structural rearrangements that precede the release of nucleotide. Regardless of the exact interpretation, it is possible that the observed structural perturbations would be even more marked in the absence of GDP. This would be in agreement with previous hydrogen–deuterium exchange experiments showing that Gs in complex with the β2AR complex in the presence of excess GDP (1 mM) undergoes changes in the exact same regions as the nucleotide-free complex but of lower magnitude (Supplementary Figs 2 and 9 of ref. 10).

The combined NMR, computational and mutagenesis analyses presented here provide a detailed view of the structural features

that govern the GBA–Gα interaction. Because some GBA proteins are specifically dysregulated in disease and disruption of Gα binding blocks their adverse effects[16,20,31], the structural insights gained here should provide a solid framework for the development of new pharmacological agents. On the basis of our results, such agents would not be expected to interfere with GPCR-mediated activation, and previous data indicate that GIV's GBA motif binding can be disrupted by mutagenesis of Gα without affecting binding to other G-protein-interacting partners like GPCRs, Gβγ, RGS proteins or GoLoco guanine-nucleotide dissociation inhibitors[25]. Although effectors bind to the SwII/α3 region of Gα, they do so only in its GTP-bound conformation. This conformation is different from that adopted by Gα-GDP[3], the form to which GBA motifs bind, thereby providing room for specificity in targeting. However, a major challenge will be to identify drugs that can discriminate among different GBA proteins. Such efforts would require atomic resolution structures to exploit small differences in the binding mode among distinct GBA proteins.

## Methods

**Preparation of nucleotide-free Gαi3 and pull-down assay.** His-tagged rat Gαi3 cloned in a pET28b vector was purified from BL21(DE3) bacteria (Life Technologies) after isopropyl-[beta]-D-thiogalactoside (IPTG) induction[18,25]. Nucleotide-free Gαi3 was prepared by promoting nucleotide release with high concentrations of ammonium sulfate in the presence of glycerol[22,50]. For this, His-Gαi3 was run through a Superdex S75 column equilibrated with 20 mM Tris-HCl (pH 7.4), 20 mM NaCl, 1 mM MgCl$_2$, 1 mM dithiothreitol (DTT), 10 μM GDP and 5% (v/v) glycerol. A volume of 200 μl of His-Gαi3 at ~5 μM were loaded on a Sephadex G-25 column (10 ml) equilibrated with 1 M ammonium sulfate, 20% (v:v) glycerol (pH 6) and the protein eluted at room temperature with the same buffer. Protein-containing fractions were pooled (~400–600 μl) and incubated overnight at 4 °C. After concentration to ~100–200 μl, the sample was applied to a second Sephadex G-25 column (10 ml) equilibrated with 50 mM Tris-HCl (pH 8.1), 150 mM NaCl, 2 mM MgCl$_2$, 1 mM EDTA, 2 mM DTT, 20% (v/v) glycerol and 0.04% (w:v) C$_{12}$E$_{10}$, and eluted with the same buffer. Protein-containing fractions were pooled and subsequently separated in three aliquots of 125 μl. One aliquot was kept at 4 °C (nucleotide-free Gαi3, and Gαi3-[ ]), the other two were incubated at 30 °C for 3 h in the presence of 100 μM GDP or 100 μM GDP plus 100 μM AlCl$_3$ and 10 mM NaF to generate GDP-loaded Gαi3 (Gαi3-[GDP]) or GDP/AlF$_4^-$-loaded Gαi3 (Gαi3-[GTP]). Samples were cooled down to 4 °C and centrifuged for 10 min at 14,000g. A volume of 30 μl of the His-Gαi3-containing supernatants were used for pull-down assays in which the proteins immobilized on glutathione–agarose beads were 10 μg of glutathione S-transferase (GST), GST-GIV (residues 1,671–1,755, containing its GBA motif)[20], GST-GAIP[25] or GST-Ric-8A (residues 12–491, a plasmid generously provided by Stephen Sprang, University of Montana)[22]. Binding reactions were carried out for 2.5 h at 4 °C in a final volume of 300 μl of 50 mM Tris-HCl (pH 8.1), 150 mM NaCl, 2 mM MgCl$_2$, 1 mM EDTA, 2 mM DTT, 20% (v:v) glycerol and 0.04% (w:v) C$_{12}$E$_{10}$, or the same buffer supplemented with 100 μM GDP or 100 μM GDP plus 100 μM AlCl$_3$ and 10 mM NaF. Glutathione–agarose beads were washed with the same buffers and resin-bound proteins eluted with Laemmli sample buffer and separated by SDS–PAGE. After transfer to polyvinylidene difluoride (PVDF) membranes, GST-fused proteins were visualized by Ponceau S staining and His-tag detected by immunoblotting with mouse anti-polyHis-Tag primary antibodies (1:2,500, Sigma, H1029) and goat anti-mouse IRDye 800F(ab')2 secondary antibodies (1:10,000, Li-Cor) using an Odyssey Infrared Imaging System (Li-Cor Biosciences). Equal volumes of the starting material of Gαi3-[ ], Gαi3-[GDP] and Gαi3-[GTP] were run to verify equivalent loading of protein in the binding reactions. Images were processed using the Image J software (NIH) and assembled for presentation using Photoshop and Illustrator softwares (Adobe). Images of uncropped scans of immunoblots and protein gels are provided in Supplementary Fig. 9.

**Co-immunoprecipitation.** HEK293T cells (American Type Culture Collection) were grown at 37 °C in DMEM supplemented with 10% FBS, 100 U ml$^{-1}$ penicillin, 100 μg ml$^{-1}$ streptomycin, 1% L-glutamine and 5% CO$_2$. DNA plasmids encoding for C-terminally 3xFLAG-tagged Gαi3 (ref. 25) or an empty plasmid (p3xFLAG-CMV-14) were transfected using the calcium phosphate method in 10 cm dishes. The DNA amount for the Gαi3 N269D plasmid was threefold (12 μg) of that for Gαi3 WT (4 μg) to equalize their expression levels. Cells were collected 48 h after transfection by gently scrapping in PBS, centrifuged and resuspended in 1 ml of lysis buffer (20 mM HEPES (pH 7.2), 5 mM Mg(CH$_3$COO)$_2$, 125 mM K(CH$_3$COO), 0.4% (v:v) Triton X-100, 1 mM DTT, 10 mM β-glycerophosphate and 0.5 mM Na$_3$VO$_4$ supplemented with a protease inhibitor cocktail (Sigma; catalogue #S8830)) and cleared (14,000g for 10 min) before use. Lysates

were supplemented with 1 μg of purified GST-Ric-8A (12–491)[22] and 2 μg of mouse anti-FLAG antibody (Sigma F1804), and incubated for 4 h at 4 °C with rotation. BSA-blocked Protein G agarose beads (Thermo-Scientific) were added and tubes incubated for additional 90 min at 4 °C. Beads were washed four times (4.3 mM Na$_2$HPO$_4$, 1.4 mM KH$_2$PO$_4$, pH 7.4, 137 mM NaCl, 2.7 mM KCl, 0.1% (v/v) Tween 20, 10 mM MgCl$_2$, 5 mM EDTA and 1 mM DTT) and immunoprecipitated proteins eluted by boiling in Laemmli sample buffer. Proteins were separated by SDS–PAGE, transferred to PVDF membranes and immunoblotted with rabbit anti-GIV serum (1:500)[51], rabbit anti-GST (1:500, Santa Cruz Biotechnology Z-5), mouse anti-FLAG (1:1,000) and mouse α-tubulin (1:2,500) primary antibodies followed by incubation with goat anti-rabbit and goat anti-mouse Alexa Fluor 680 or IRDye 800F(ab')2 secondary antibodies (1:10,000) and imaging with an Odyssey Infrared Imaging System. Images were processed using the Image J software (NIH) and assembled for presentation using Photoshop and Illustrator software (Adobe).

**Purification of proteins and synthesis of peptides for NMR.** Full-length human Gαi3 (UNIPROT entry P08754) cloned in a pET24d(+) plasmid (a generous gift from Ichio Shimada, University of Tokyo) was used to purify the protein used in NMR experiments[39,47]. For this, protein expression was induced in BL21 Rosetta (Life Technologies) cell cultures with 1 mM IPTG for 16 h at 23 °C in a modified M9 minimal medium containing 1 g l$^{-1}$ $^{15}$N-NH$_4$Cl, 2 g l$^{-1}$ $^2$H-$^{13}$C-glucose and 1 g l$^{-1}$ of $^2$H-$^{13}$C-$^{15}$N Celtone base powder in $^2$H$_2$O with 30 mg l$^{-1}$ kanamycin and 34 mg l$^{-1}$ chloramphenicol. The pelleted cells were lysed by sonication in 50 mM Tris, 200 mM NaCl, 100 mM KCl and 1 mM DTT at pH 8.0, and the supernatant was loaded on a Ni$^{2+}$-loaded His-Trap FF column. The fractions were eluted with the same buffer plus 20 μM GDP and 500 mM imidazole, and the N-terminal His-tag was cleaved with a GST-fusion of the HVRV3C protease (using a 8.3:1 mass ratio) during an overnight dialysis at 4 C to remove imidazole. This sample was loaded again on a His-Trap column and the unbound fraction was loaded on a GST-Trap column to remove the protease. The flow-through was concentrated by ultrafiltration and further purified by gel filtration on a Superdex 200 in 10 mM HEPES, 150 mM NaCl, 1 mM DTT and 20 μM GDP at pH 7.0. Protein integrity was verified by mass spectrometry. Between 5 and 9 mg of protein per litre of culture were obtained.

His-tagged human GIV residues 1,660–1,870 (GIV-CT; residue numbers refer to the sequence in UNIPROT entry Q3V6T2-3) cloned in pET28b (ref. 25) was expressed in Escherichia coli BL21(DE3) by isopropyl-β-D-thiogalactoside induction. The pellet of a 30 ml saturated culture of BL21 cells in Luria-Bertani medium was used to inoculate 1 l of an autoinducible ZYP-5052 medium[52] with 30 mg l$^{-1}$ kanamycin. Cultures were grown for 2 h at 37 °C and then for 16 h at 20 °C. Initial tests indicated that only a small fraction of the induced protein was soluble. To improve the efficiency of the immobilized metal ion chromatography purification of the soluble protein, the periplasmic material was removed before cell lysis[53]. The pelleted cells of 6 l of bacterial culture were then sonicated in 50 mM sodium phosphate (pH 7.4) with 300 mM NaCl and protease inhibitors. The soluble protein was captured on a 5 ml HiTrap Chelating HP column loaded with Co$^{2+}$ ions in the same buffer and eluted with a gradient from 20 to 500 mM imidazole in 40 column volumes. Protein-containing fractions were pooled and imidazole was removed (by means of a HiPrep Desalting column) before injection in a 1 ml HiTrap column loaded with Ni$^{2+}$ ions. On protein elution using the same gradient as in the previous chromatography, two overlapping peaks corresponding to two proteins with a small difference in apparent molecular weight (as seen in SDS–PAGE) were pooled and the proteins were further separated on a Superdex 200 16/60 column run with 20 mM sodium phosphate (pH 7.4), 500 mM NaCl and 1 mM DTT. By mass spectrometry peptide fingerprinting, we identified the protein eluting at 78 ml as GIV, and MALDI was consistent with the calculated mass of the CT fragment with the N-terminal His-tag but lacking the initial methionine. The yield was 0.2 mg of pure protein per litre of culture. The uniformly $^{15}$N-labelled protein was produced in the same way but growing the cells in autoinducible medium containing 1.3 g l$^{-1}$ $^{15}$N-NH$_4$Cl.

A peptide corresponding to GIV amino acids 1,671–1,696 (KTGSPGSEVVTLQQFLEESNKLTSVQ) was synthesized using the in situ neutralization protocol for Boc-Solid Phase Peptide Synthesis[54] on a p-methylbenz-hydrylamine resin (Novabiochem, 0.67 mmol g$^{-1}$, 100–200 mesh). Following chain elongation, the peptide was cleaved from the resin using a solution of hydrofluoric acid containing a 5% of anisole for 1 h at 0 °C. Next, the solution was removed under vacuum and the resulting residue crushed out with Et$_2$O and filtered. The collected solid was redissolved in a 50% CH$_3$CN/H$_2$O solution containing 0.1% of trifluoroacetic acid (TFA), frozen down and lyophilized. The crude peptide was purified by reverse phase-HPLC in a Waters X-Bridge C18 (19 × 100 mm) column at a flow of 20 ml min$^{-1}$ using H$_2$O (0.1% TFA) and CH$_3$CN (0.1% TFA) as eluents. The identity and final purity (>97%) of the peptide was determined by analytical reversed phase HPLC and mass spectrometry (electrospray ionization-time of flight, ESI-TOF).

**NMR spectroscopy.** All NMR data were measured on a Bruker Avance III 800 MHz (18.8 T) spectrometer equipped with a cryogenically cooled triple resonance z-gradient probe, processed with Topspin (Bruker) and analysed with Sparky. Proton chemical shifts were referenced to internal 2,2-dimethyl-2-

silapentane-5-sulfonate (DSS, 0.00 p.p.m.), and $^{13}$C and $^{15}$N chemical shifts were indirectly referenced to DSS[55]. Gαi3 spectra were recorded at 30 °C on $^2$H-$^{13}$C-$^{15}$N–Gαi3 samples with a threefold molar excess of GDP in 10 mM HEPES (pH 7.0) with 10 mM MgCl$_2$, 5 mm DTT, 0.01% NaN$_3$ and 5% $^2$H$_2$O. The samples were prepared by mixing different amounts of Gαi3 and GIV-CT from stocks in the corresponding gel filtration buffers followed by three cycles of fourfold dilution in the NMR buffer and concentration by ultrafiltration using 10 kDa cut-off membranes. Protein solubility limited the Gαi3 concentration of the samples to 100 μM (free and bound to GIV-CT) or 32 μM (bound to the 1,671–1,696 GIV peptide). $^1$H–$^{15}$N TROSY and HNCO spectra of Gαi3 allowed transferring most of the assignments of the protein backbone resonances deposited in the BiomagResDataBase entry 19015. We observed, however, a systematic offset of 0.09 and −1.1 p.p.m. in our spectra with respect to the published $^1$H and $^{15}$N chemical shifts, respectively. The assignment of Gαi3 resonances in the presence of a twofold molar excess of GIV-CT was done based on $^1$H–$^{15}$N TROSY, HNCO and HNcoCA spectra (the latter one, however, lacking many signals because of its lower sensitivity). The assignment of Gαi3 resonances in the presence of a fivefold molar excess of GIV fragment 1,671–1,696 was achieved from the joint analysis of an HNCO and $^1$H–$^{15}$N TROSY spectra recorded along a titration (at molar ratios 1:0, 1:0.2,1:0.5, 1:1, 1:2 and 1:5). The stepwise addition of the fragment from a 5 mM stock in NMR buffer with 50% $^2$H-dimethylsulfoxide (DMSO; necessary to solubilize the peptide) resulted in a 3% DMSO concentration at the last point of the titration. On the basis of numerous observations on other proteins, we assume that this concentration of DMSO is too low to significantly affect the chemical shifts of protein amide protons[56]. The CSPs were computed as the weighted average distance between the backbone amide $^1$H and $^{15}$N chemical shifts in the free and bound states[57]. To compare the intensity of Gαi3 signals in different TROSY spectra, the absolute values were divided by the intensity of the C-terminal Y354, a narrow isolated signal and one of the most intense, which remained essentially unchanged on GIV binding. The intensity ratios ($I_{ratio}$) for each signal were calculated by dividing the normalized intensity in the free form ($I_{free}$) by the normalized intensity in the bound form ($I_{bound}$). For those signals in which the $I_{bound}$ was <1% of the $I_{free}$, it was treated as 1% for all subsequent calculations and representations (that is, $I_{ratio} = 100$, $\log_{10}(I_{ratio}) = 2$).

GIV-CT $^1$H–$^{15}$N HSQC spectra were recorded at 25 °C on a 125 μM sample in PBS (pH 5.5) with 0.5 mM DTT, 5% $^2$H$_2$O and 20 μM DSS.

**Circular dichroism.** Circular dichroism measurements were performed on a Jasco J-810 spectropolarimeter (JASCO, Tokyo, Japan). The circular dichroism spectrum was recorded on a GIV-CT protein sample 8.4 μM in PBS (pH 7.0) with 0.2 mM DTT in a 0.2 cm path length quartz cuvette at 25 °C. The thermal denaturation curve from 5 to 95 °C was recorded on the same protein sample and cuvette by increasing temperature at a rate of 1 °C min$^{-1}$ and measuring the change in ellipticity at 222 nm.

**Modelling of the GIV–Gαi3 complex.** A model of human Gαi3 bound to the GIV GBA motif (residues 1,678–1,696) was generated by homology in combination with iterative stages of ab initio folding, protein–protein docking and refinement. Independent models of Gαi3 and GIV–GBA (residues 1,678–1,688) were first built via homology from the X-ray crystal structure of human Gαi1 bound to the KB-752 synthetic GEF peptide using ICM version 3.8-3 (Molsoft LLC., San Diego, CA). The human Gαi3 (NCBI accession: P08754) and GIV (NCBI accession: BAE44387) sequences were aligned to those of Gαi1 or the KB-752 peptide and separately threaded through the corresponding 3D subunit structures of the Gαi1:KB752 complex (PDB: 1Y3A, chains B and F, respectively). Before model building, hydrogen was added to the 1Y3A parent structure and the isomeric/tautomeric state and orientation of side chains was optimized with ICM to find the best global hydrogen bond networks and local energetics. Subsequent nonconserved side-chain conformation optimization and loop prediction (residues 112–117 on Gαi3) for the model were performed by the biased probability Monte Carlo ICM method[58–60], which refines the global energy with respect to ECEPP/3 and solvation energy terms. N- and C-terminal extensions for Gαi3 beyond the alignment coverage were modelled de novo by extending the existing structure as helical segments then minimizing side chains as above.

To model the Gαi3:GIV interaction, protein–protein docking of the GIV–GBA peptide (residues 1,678–1,688) and Gαi3 models was performed with ICM using a rigid-body two-stage fast Fourier transform method followed by flexible refinement of ligand/receptor interface residues[61]. To extend the model for inclusion of GIV residues 1,678–1,696, eight additional amino acids were added to the C terminus of the top-scoring GIV–GBA conformation from the docking solutions above. The residues added to the complex were populated in a standard starting covalent geometry with free torsion angles, and their Gαi3-engaging conformation was simulated ab initio with ICM using the biased probability Monte Carlo-based approach. Simulations were conducted within a rigid-receptor context at 300 K in continuous dielectric solvent during which GIV residues 1,678–1,687 were tethered to their protein–protein docking solution while the backbone and side chains of the extended residues were freely flexible. The extended GIV–GBA peptide was removed from the receptor and re-docked as above to confirm that an energetically favourable solution had been found. A fragment-guided molecular dynamics simulation was then used to improve the modelled complex's local geometry by relaxing steric

strains and to optimize torsion angles and hydrogen-bonding networks[62]. FoldX version 3.0 RepairPDB feature was next used to identify and repair any remaining high-energy side-chain conformations[63,64]. Per-residue energy contributions (kcal mol$^{-1}$) to the stability of the modelled complex were estimated with FoldX-SequenceDetail by calculating the differences (ΔΔG) between the GIV-bound complex and monomeric Gαi3 residues at pH 7.0, 0.05 M ionic strength and 298 K. Those residues that remained > + 1 s.d. in the free energy change (ΔΔG) between the Gαi3 monomer and the Gαi3:GIV complex were considered to be strained, and their side chains were individually optimized with respect to their local environment as a final round of refinement with ICM as described above. GDP was placed within the final model based on the 1Y3A coordinates and neighbouring side chains surrounding the ligand were optimized. Model images were generated with PyMOL Molecular Graphics System (Schrödinger, LLC).

**Purification of Gαi3 mutants for biochemical studies.** Human Gαi3 was expressed from the same plasmid as described for the NMR studies and purified as an uncleaved His-tagged protein. Pelleted bacteria from 1 l of culture were resuspended in 25 ml of buffer (50 mM NaH$_2$PO$_4$, pH 7.4, 300 mM NaCl, 10 mM imidazole, 25 μM GDP and 1% (v:v) Triton X-100 supplemented with protease inhibitor cocktail (leupeptin 1 μM, pepstatin 2.5 μM, aprotinin 0.2 μM and phenylmethylsulfonyl fluoride 1 mM)). After sonication (four cycles, with pulses lasting 20 s and with 1 min interval between pulses to prevent heating), lysates were centrifuged at 12,000g for 20 min at 4 °C. Solubilized proteins were affinity-purified on HisPur Cobalt Resin (Pierce) and eluted with lysis buffer supplemented with 250 mM imidazole. The buffer was exchanged for 20 mM Tris-HCl, pH 7.4, 20 mM NaCl, 1 mM MgCl$_2$, 1 mM DTT, 10 μM GDP and 5% (v/v) glycerol using a HiTrap Desalting column (GE Healthcare). Protein samples were stored at −80 °C.

**Gαi3-limited proteolysis assay.** Human His-Gαi3 (0.25 mg ml$^{-1}$) was incubated for 150 min at 30 °C in buffer (20 mM Na-HEPES, pH 8, 100 mM NaCl, 1 mM EDTA, 10 mM MgCl$_2$, 1 mM DTT and 0.05% (w:v) C$_{12}$E$_{10}$) supplemented with GDP (30 μM) or GTPγS (30 μM). After incubation trypsin was added to the tubes (final concentration 6.25 μg ml$^{-1}$) and samples were incubated for 10 min at 30 °C. Samples were rapidly transferred to ice and reactions stopped by the addition of Laemmli sample buffer and boiling for 5 min. Proteins were separated by SDS–PAGE and stained with Coomassie blue.

**In vitro protein-binding assays with GST-fused proteins.** Binding of Gαi3 mutants to GST-fused GIV or DAPLE was determined by a pull-down assay[19,25]. A unit of 10 μg of GST, GST-GIV (1,671–1,755)[20] or GST-DAPLE (1,650–1,745, created by cloning from Kazusa's clone fh14721 (KIAA1509) into pGEX-4 T-1) were immobilized on glutathione agarose beads for 90 min at room temperature in PBS (∼1.25 μM final concentration in the binding reactions). Beads were washed twice with PBS, resuspended in 250 μl of binding buffer (50 mM Tris-HCl, pH 7.4, 100 mM NaCl, 0.4% (v:v) NP-40, 10 mM MgCl$_2$, 5 mM EDTA, 2 mM DTT and 30 μM GDP) and incubated 4 h at 4 °C with constant rotation in the presence of His-tagged Gαi3 WT or mutants (∼1 μM final concentration). Beads were washed four times with 1 ml of wash buffer (4.3 mM Na$_2$HPO$_4$, 1.4 mM KH$_2$PO$_4$, pH 7.4, 137 mM NaCl, 2.7 mM KCl, 0.1% (v/v) Tween-20, 10 mM MgCl$_2$, 5 mM EDTA, 1 mM DTT and 30 μM GDP) and resin-bound proteins eluted with Laemmli sample buffer by incubation at 37 °C for 10 min. Proteins were separated by SDS–PAGE and stained with Coomassie blue.

**Fluorescence polarization-based peptide-binding assays.** Fluorescently labelled peptides derived from human GIV (residues 1,671–17,01, KTGSPGSEVVTLQ QFLEESNKLTSVQIKSSS), DAPLE (residues 1,662–1,695, SASPSSEMVTLEEF LEESNRSSPTHDTPSCRDDL) or RGS12 (residues 1,185–1,221, DEAEEFFELIS KAQSNRADDQRGLLRKEDLVLPEFLR) were synthesized following a protocol as described above in 'Purification of proteins and synthesis of peptides for NMR' with minor modifications. Briefly, following chain elongation 5,6-carboxy-fluorescein was activated with HATU and coupled to the resin-bound peptide at 65 °C for 1 h to yield the fluorescein-labelled peptides. The remaining steps were carried out as described above. Fluorescence polarization measurements were carried out in 384-well plates (Black OptiPlate-384F, Perkin Elmer). G protein (0–8 μM) and peptide (0.025 μM) were mixed at room temperature for 10 min in a final volume of 20 μl of binding buffer (50 mM Tris-HCl, pH 7.4, 100 mM NaCl, 0.4% (v:v) NP-40, 10 mM MgCl$_2$, 5 mM EDTA, 2 mM DTT and 30 μM GDP). Fluorescence polarization (excitation 485 ± 10 nm/emission 528 ± 10 nm) was measured every 2 min for 30 min at room temperature in a Biotek H1 synergy plate reader to ensure that the signals were stable in time. Fluorescence polarization at different times was averaged, normalized and fitted to a one site binding hyperbola to determine the equilibrium dissociation constant ($K_d$). For mutants not reaching binding saturation, the maximal binding of Gαi3 WT measured in the same experimental set was considered 100% binding for normalization.

**GTPase and GTPγS-binding assays with GIV and DAPLE.** Steady-state GTPase activity was measured by release of radioactive phosphate and GTPγS binding by

retention of radioactive nucleotide on filters[19,25]. Human His-Gαi3 WT or mutants (100 nM final concentration) were pre-incubated with His-GIV-CT (residues 1,660–1,870, ref. 25), 3 μM final concentration) or a DAPLE-derived peptide (residues 1,662–1,695, SASPSSEMVTLEEFLEESNRSSPTHDTPSCRDDL, 30 μM final concentration) for 15 min at 30 °C in assay buffer (20 mM Na-HEPES, pH 8, 100 mM NaCl, 1 mM EDTA, 25 mM MgCl$_2$ and 0.05% (w:v) C$_{12}$E$_{10}$). GTPase and GTPγS-binding reactions were initiated at 30 °C by adding an equal volume of assay buffer containing 1 μM [γ-$^{32}$P]GTP (~50 c.p.m. fmol$^{-1}$) or 1 μM [$^{35}$S] GTPγS (~50 c.p.m. fmol$^{-1}$), respectively. GTPase reactions were stopped at 15 min in duplicates by mixing 25 μl of the reaction with 975 μl of ice-cold 5% (w/v) activated charcoal in 20 mM H$_3$PO$_4$, pH 3. Samples were then centrifuged for 10 min at 10,000g, and 500 μl of the resultant supernatant were scintillation-counted to quantify the amount of [$^{32}$P]Pi released. Background [$^{32}$P]Pi detected at 15 min in the absence of G protein was subtracted from each reaction (<5% of the counts detected in the presence of G proteins). GTPγS-binding reactions were stopped at 15 min in duplicates by addition of 3 ml, ice-cold wash buffer (20 mM Tris-HCl, pH 8.0, 100 mM NaCl and 25 mM MgCl$_2$) followed by rapid filtration through BA-85 nitrocellulose filters (GE Healthcare). Filters were washed with 4 ml of the same buffer, dried and subjected to liquid scintillation counting. Background [$^{35}$S]GTPγS binding to the filters at 15 min in the absence of G protein was subtracted from each reaction (<5% of the counts detected in the presence of G proteins). GTPase activity and GTPγS binding at 15 min are within the linear region of the initial rate of the reactions[19,25]. For both assays, GEF-mediated activation was expressed as per cent of the basal activity (GTPase hydrolysis or GTPγS binding) of each His-Gαi3 protein in the absence of GIV or DAPLE.

**Synthesis of peptide libraries and Gαi3 overlay.** Libraries of immobilized peptides were produced by automatic SPOT synthesis on continuous cellulose membrane supports (Whatman 50 cellulose membranes) using Fmoc (fluoren-9-ylmethoxycarbonyl) chemistry with the AutoSpot-Robot ASS 222 (Intavis Bioanalytical Instruments AG)[65]. Individual peptide–cellulose complexes were solubilized and re-spotted on Celluspot slides for subsequent probing. Slides were primed in binding buffer (4.3 mM Na$_2$HPO$_4$, 1.4 mM KH$_2$PO$_4$, pH 7.4, 137 mM NaCl, 2.7 mM KCl, 5 mM MgCl$_2$, 1 mM DTT, 30 μM GDP and 1% (v:v) TX100) and blocked for 1 h in the same buffer supplemented with 5% (w:v) BSA. Slides were incubated for 2 h at room temperature with rat His-Gαi3 at 20 μg ml$^{-1}$ (~0.5 μM) in the same buffer. After four washes, slides were sequentially incubated with primary (rabbit anti-Gαi3, 1: 250; 90 min) and secondary (goat anti-rabbit Alexa Fluor 680, 1:10,000; 60 min) antibodies. Images were acquired in an Odyssey infrared scanner (Li-Cor), processed using the Image J software (NIH) and assembled for presentation using Photoshop and Illustrator softwares (Adobe).

**Gβγ-binding assays.** Gβγ was purified from bovine retinas, which are primarily composed of Gβ$_1$γ$_1$ by purification of holotransducin from rod outer segment (ROS) membranes isolated from dark-adapted bovine followed by separation of the Gβ$_1$γ$_1$ complex from the α-subunit of transducin on a Hitrap Blue-sepharose column (GE healthcare), and further purification by anion exchange chromatography using a Hitrap-Q column[66,67].

For binding assays, Gβ$_1$γ$_1$ (100 nM final) was incubated in the absence or presence of human His-tagged Gαi3 WT or mutants (50 nM final) in 250 μl of binding buffer (50 mM Tris-HCl, pH 7.4, 100 mM NaCl, 0.4% (v:v) NP-40, 5 mM MgCl$_2$, 10 mM imidazole, 0.1 mg ml$^{-1}$ BSA, 1 mM DTT and 30 μM GDP) for 4 h at 4 °C with rotation. A volume of 25 μl of BSA-blocked HisPur cobalt resin (Pierce) were added to each tube and the incubation continued for 90 min. Beads were washed four times with 1 ml of binding buffer and resin-bound proteins eluted by adding Laemmli sample buffer supplemented with 100 mM EDTA and boiling for 5 min. Proteins were separated by SDS–PAGE, transferred to PVDF membranes and immunoblotted with rabbit anti-Gαi3 (1:500, Santa Cruz Biotechnology C-10) and anti-panGβ (1:250, Santa Cruz Biotechnology M-14) primary antibodies followed by incubation with goat anti-rabbit Alexa Fluor 680 (1:10,000, Life Technologies) secondary antibodies and imaging with an Odyssey Infrared Imaging System (Li-Cor Biosciences).

**Steady-state GTPase assays with rhodopsin.** ROS membranes washed with urea were prepared as described earlier as the source of rhodopsin[67]. Human His-tagged Gαi3 WT and mutants (20–25 μM) were incubated overnight at 4 °C with twofold excess Gβ$_1$γ$_1$ in 20 mM Tris-HCl, pH 7.4, 20 mM NaCl, 1 mM MgCl$_2$, 1 mM DTT, 10 μM GDP and 5% (v/v) glycerol. Bovine rhodopsin was freshly solubilized from urea-washed ROS membranes in assay buffer (20 mM HEPES, pH 7.5, 100 mM NaCl, 5 mM MgCl$_2$, 1 mM DTT and 0.01% (w:v) dodecylmaltoside) for each experiment. Gαi3/Gβ$_1$γ$_1$ (100/200 nM final) and rhodopsin (20 nM final) were incubated in ice for 30 min under ambient light in assay buffer. Reactions were initiated at 30 °C by adding an equal volume of assay buffer containing 1 μM [γ-$^{32}$P]GTP (~50 c.p.m. fmol$^{-1}$) and stopped in duplicates by mixing 25 μl of the reaction with 975 μl of ice-cold 5% (w/v) activated charcoal in 20 mM H$_3$PO$_4$, pH 3. Samples were then centrifuged for 10 min at 10,000g, and 500 μl of the resultant supernatant were scintillation-counted to quantify the amount of

[$^{32}$P]Pi released. Background [$^{32}$P]Pi detected at 15 min in the absence of G protein was subtracted from each reaction (<5% of the counts detected in the presence of G proteins). Results were expressed as per cent of the activity of Gi3 WT without rhodopsin.

**G-protein activation assay in yeast.** The previously described S. cerevisiae strain CY1316 (ref. 68; MATα gpa1Δ far1Δ fus1p-HIS3 ste14:trp1:LYS2 ste3Δ lys2 ura3 leu2 trp1 his3 can1; kindly provided by James Broach, Penn State University) was used for experiments with GIV and DAPLE and strain MMY11 (ref. 69; MATa gpa1Δ far1Δ fus1p-HIS3 ste2Δ sst2Δ FUS1-lacZ ura3 trp1 his3 can1; kindly provided by Simon Dowell, Glaxo-Smith-Kline) for the experiments with A2b receptor. The native G-protein-dependent pheromone response pathway is similarly modified in both strains: the pheromone-activated GPCR (Ste2 or Ste3), endogenous Gα-subunit (Gpa1) and the cell cycle arrest-inducing protein far1 are deleted. In MMY11, the G-protein inhibitor Sst2 is also deleted. Both strains were transformed with a centromeric plasmid (CEN TRP) encoding a chimeric yeast Gpa1(1–41)-human Gαi3 (36–354) protein under the control of the endogenous Gpa1 promoter[68] (courtesy of Mary Cismowski, Nationwide Children's Hospital) using the lithium acetate method[70]. In these strains, the pheromone response pathway can be upregulated by the ectopic expression of activators of human Gαi3 and does not result in the cell cycle arrest that occurs in the native pheromone response[68,69]. CY1316-derived strains were transformed with pYES2 plasmids (2 μm, URA3) encoding for GIV (residues 1,660–1,870, between BglII/EcoRI) or DAPLE (residues 1,650–1,880, between BamHI/NotI with an N-terminal myc-tag) and double transformants selected in SD-Trp-Ura media. Individual colonies were inoculated into 3 ml of SDGalactose-Trp-Ura and incubated overnight at 30 °C to induce the expression of the proteins of interest under the control of the galactose-inducible promoter of pYES2. This starting culture was used to inoculate 20 ml of SDGalactose-Trp-Ura at 0.3 OD600. Exponentially growing cells (~0.7–0.8 OD600, 4–5 h) were pelleted to prepare samples for immunoblotting (see below). MMY11-derived strains were transformed with pDT-PGK plasmids (2 μm, URA3) encoding for human A2bR WT and A2bR G135A/I197L/Y202N (courtesy of Ad P. IJzerman, Leiden University), a mutant that constitutively activate G proteins in the absence of agonist[71]. These strains were processed as described above except that the media contained glucose instead of galactose.

Yeast protein samples for immunoblotting were prepared by acid lysis and precipitation followed by neutralization and solubilization[70]. Pellets corresponding to 5 OD600 were washed once with PBS + 0.1% BSA and then resuspended in 150 μl of lysis buffer (10 mM Tris-HCl, pH 8.0, 10% (w:v) trichloroacetic acid, 25 mM NH$_4$OAc and 1 mM EDTA). A volume of 100 μl of glass beads was added to each tube and samples were vortexed at 4 °C for 5 min. Lysates were separated from the glass beads by poking a hole in the bottom of the tubes followed by centrifugation into new tubes. The procedure was repeated after adding 50 μl of lysis buffer to wash the glass beads. Proteins were then precipitated by centrifugation (10 min, 20,000g) and resuspended in 60 μl of solubilization buffer (0.1 M Tris-HCl, pH 11.0, and 3% SDS). Samples were boiled for 5 min and centrifuged for 1 min at 20,000g. A volume of 50 μl from the supernatant was transferred to new tubes containing 12.5 μl of Laemmli sample buffer and boiled for 5 min. Proteins (~15–20 μl per lane) were separated by SDS–PAGE, blocked in PBS supplemented with 5% BSA and analysed by sequential incubation with primary and secondary antibodies. Primary antibodies were diluted as follows: ppERK (rabbit mAb, Cell Signaling #4370), which recognizes yeast ppFus3 1:2,500; myc (mouse mAb, Cell Signaling #9B11): 1:1,000, Girdin (Santa Cruz Biotechnology, T-13); and α-tubulin (Sigma T6074): 1:2,500. Secondary antibodies (goat anti-mouse IRDye 800F(ab′)2, Li-Cor Biosciences, and goat anti-rabbit Alexa Fluor 680, Life Technologies) were used at 1:10,000 dilution. Images were acquired in an Odyssey infrared scanner (Li-Cor), processed with the Image J software (NIH) and assembled for presentation using Photoshop and Illustrator softwares (Adobe).

Out of the five Gαi3 mutants tested in the in vitro G-protein activation experiments with rhodopsin (K35A, L249H, S252D, N256E and ΔC9), only N256E and ΔC9 were tested for different reasons. K35A was excluded because this residue is right at the boundary between the Gαi3 and Gpa1 sequences of the chimera and for this reason it may yield results that are difficult to interpret. L249H and S252D were excluded because they displayed enhanced spontaneous activation in yeast.

**Data availability.** The UniProt accession codes P08754 and Q3V6T2-3, PDB accession codes 1Y3A and 1GDD, and BMRB accession code 19015 were used in this study. The coordinates of the final model have been deposited in the open access repository Model Archive (www.modelarchive.org) (10.5452/ma-ayq5v). All other data are available from the corresponding authors on reasonable request.

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

## Acknowledgements

This work was supported by NIH grants R01GM108733 and R01GM112631, American Cancer Society grants RSG-13-362-01-TBE and IRG-72-001-36, and the Karin Grunebaum Foundation (to M.G.-M.) and CTQ2014-56966-R (to F.J.B.). V.D. is a recipient of a postdoctoral fellowship from the Hartwell Foundation. We thank Ichio Shimada (University of Tokyo), Mary Cismowski (Nationwide Children's Hospital) and Ad P. IJzerman (Leiden University) for providing plasmids and Simon Dowell (Glaxo-Smith-Kline) and James Broach (Penn State University) for providing yeast strains. We thank S. Sprang and C. Thomas (U. of Montana) for providing plasmids and detailed protocols for the preparation of nucleotide-depleted Gαi. We thank Jane Findlay and Sunitha Yelleswarapu for technical support.

## Author contributions

A.I.d.O. performed and analysed NMR experiments; N.M. and M.V. purified proteins for NMR studies and performed CD experiments; K.P.-S., V.D., A.L., A.M., L.T.N. and M.G.-M. performed protein-binding experiments, yeast-based assays, G-protein activity experiments and trypsin protection assays; V.D. conducted the computational modelling and analysed data; M.A.d.l.C.-M. performed the co-immunoprecipitation experiments; J.B.B.-C. synthesized peptides in solution and G.S.B. provided the peptide arrays; S.R. and R.A.C. isolated ROS membranes and purified retinal Gβγ; M.G.-M. conceived the study; M.G.-M and F.J.B. co-supervised the study, designed experiments and analysed data; M.G.-M, V.D. and F.J.B. wrote the manuscript with input from all the authors.
