## [Peer review file · Nature Communications]

Reviewers' comments:

Reviewer #1 (Remarks to the Author):

In their paper entitled “Molecular mechanism of Gai activation by non-GPCR proteins with a Ga-Binding and Activating (GBA) motif” the authors describe the interaction between the mechanism of activation of G-alpha by non-GPCR proteins via an GBA motif. They employ NMR spectroscopy and site directed mutagenesis to map this interaction in a residue resolution.

This paper provides important insight into the mechanism of GBA-protein action. This paper is well-written and compelling in terms of data quality. The overall conclusions are justified and the appropriate literature has been cited. Therefore, I recommend publication of this paper after my minor points have been addressed properly.

Specific comments:

- 1.) On page 3, the authors describe the preparation of nucleotide-free G-alpha. Has the absence of nucleotides been probed in some way, e.g. by ¹H NMR or an HPLC assay?
- 2.) The authors mention a recent crystal structure between G-alpha and a 16-mer GEF peptide (pg. 4), that is similar in sequence as the one investigated in this study. In this light, what is the novel insight obtained in the present study compared to this earlier one? The NMR data show some CSPs in regions remote from the primary binding site, but an in depth structural characterization of this feature would require by far more effort than what has been presented here. And how reliable is the molecular modeling approach taken by the authors to visualize these potential conformational changes?
- 3.) On pg. 8, the authors mention the interaction between a GPCR and G-alpha and discuss possible allosteric structural changes. A recent paper on G-alpha dynamics upon nucleotide and GPCR binding addresses this issue in detail and should be cited at this point (Goricancec et al, PNAS, 2016).
- 4.) If the GBA motif in GIV is the main interaction site, what about the other 1100 aa in the protein? Is there any evidence of the relevance of this protein in mediating protein-protein interactions or regulation of its structure and interactions by PTMs? Is there any information on the domain structure of GIV?
- 5.) It seems surprising that binding of RGS proteins like the RGS-14 GOLOCO domain is not affected by disruption of the GIV protein interface on G-alpha. At least the binding groove on the G-alpha Ras domain is extremely similar to the GIV GBA binding site that has been identified here. Maybe a structural overlay of the Ga-GOLOCO and Ga-KB-752 structures can clarify this issue.
- 6.) Page 12, third paragraph: typo: “... binding pocket I our NMR ...” should be “... in our NMR ...” and on page 13, “conformed” should be “confirmed”.

Reviewer #2 (Remarks to the Author):

This manuscript provides an in depth structure-function analysis of GIV and DAPLE, two proteins of the GBA family that activate G proteins, but which are not G protein coupled receptors. There are only a handful of non-receptor activators of G proteins. Other examples include Ric-8A and Ric-8B in higher eukaryotes and Arr4 in yeast.

The approach includes sophisticated NMR, computational modeling, mutagenesis and protein biochemistry. Experiments are done with fragments (210 aa) of GIV rather than full length protein (1870 aa). This appears to be unavoidable and the authors argue persuasively that it is a valid alternative. New mutations are described that selectively uncouple Galpha from GIV or from the receptor, and these provide supporting evidence for the mode of action suggested by the NMR data. The mutations described have the potential to reveal the relative physiological significance of the two inputs to Galpha activation.

The overall conclusion, supported by the data, is that these GBA proteins share a common and specific site of binding to G proteins, and that there are fundamental differences in GBA-mediated and receptor-mediated G protein activation (Ric8 mediated activation is distinct from the other two mechanisms).

The paper is well written. The data are described appropriately, and interpreted conservatively. Overall I am enthusiastic about the submission.

Concerns (in approximate order of importance):

The NMR analysis is the strength of the paper, but there is no comparable NMR data for Galpha when bound to GDP and receptor. There is a crystal structure of Galpha bound to a receptor, but it is static, contains two additional G protein subunits, and is in the nucleotide-free state. Thus it is somewhat of an apples-to-oranges comparison. This should be clarified in the text.

A 3D structural model of the GBA motif of GIV (residues 1678-1696) bound to Galpha is based on homology with the KB-752/Galpha structure. The authors note sequence similarities between GIV and KB-752. Given the similar structures of Galpha-GDP plus or minus KB-752, how confident are these authors about the similarity on which the homology model is based? Is it feasible to do the NMR on Galpha plus KB-752, allowing a more valid comparison?

Mutations that uncouple Galpha from GIV or from the receptor are not tested in cells or in vivo. This is probably beyond the scope of the paper, but worth mentioning as a goal for the future.

Should the peptide array data include a scrambled peptide negative control?

Use of the word "affinity" is unwarranted in places where binding is not quantified in any way (pull downs for example, vs. NMR binding analysis).

Please defend the statement: "GBA proteins bind monomeric Galpha and do not require

accessory

proteins, as opposed to GPCRs, which require Gbeta/gamma.” – What is the relative exchange rate of Galpha alone coupled to a GPCR vs. GBA? What is the relative exchange rate of Galpha/beta/gamma coupled to a GPCR vs. GBA? If these data are unpublished (refs 30 and 32) they should be included or supported with published data instead.

Please proof read the document carefully (“undergoes changes in the exact same regions”)

Reviewer #3 (Remarks to the Author):

The authors present a very complete characterization of the interaction between G-protein and GAB proteins. The combination of computational and experimental approaches allowed also to characterize the conformational changes and mechanisms underlying this interaction.

The manuscript is well written and the figures although they are a lot, are needed to comprehensively follow the article.

Minor comments:

In the methods section, particularly the computational part need more details regarding the methods, programs and algorithms used. With the data that the authors are providing, I think it is hard to reproduce their calculations.

RESPONSE TO REVIEWERS

All our responses are in boldface font following a Q&A format.

Reviewer #1

General comments: *In their paper entitled “Molecular mechanism of Gai activation by non-GPCR proteins with a Ga-Binding and Activating (GBA) motif” the authors describe the interaction between the mechanism of activation of G-alpha by non-GPCR proteins via an GBA motif. They employ NMR spectroscopy and site directed mutagenesis to map this interaction in a residue resolution.*

This paper provides important insight into the mechanism of GBA-protein action. This paper is well-written and compelling in terms of data quality. The overall conclusions are justified and the appropriate literature has been cited. Therefore, I recommend publication of this paper after my minor points have been addressed properly.

Response to General Comments- We appreciate the positive comments from the reviewer and we are glad that s/he found it suitable for publication after we address some minor points. Below we provide responses for these minor points.

Specific comments:

1.) On page 3, the authors describe the preparation of nucleotide-free G-alpha. Has the absence of nucleotides been probed in some way, e.g. by 1H NMR or an HPLC assay?

Response to Comment #1- The absence of nucleotide was probed functionally (Fig. 1B) because of limitations in the amount of material necessary to analyze it by ¹H NMR or HPLC. We purified nucleotide-free Gα following closely what, to our knowledge, is the only existing protocol for this purpose, which yields a G protein ~90% devoid of nucleotide as determined by indirect methods (see Ref.50 in the manuscript and references therein). However, this protocol only permits the preparation of small quantities of protein (~10 μg) and nucleotide-free Gα denatures at concentrations >0.2 mg/ml. This scarce amount of material precludes reliable nucleotide content measurements by ¹H NMR or HPLC.

To functionally probe for the nucleotide depletion, we monitored binding to Ric-8A. It has been previously established that Ric-8A preferentially binds nucleotide-free Gα, while the presence of nucleotides in excess leads to complex dissociation. Thus, our results in Fig. 1B indicate that nucleotide depletion of Gα was efficient because Ric-8A bound robustly to Gα after the nucleotide depletion procedure and because this binding was dramatically reduced (~10 fold or higher) upon nucleotide addition. These observations, together with the previously reported 90% depletion using an almost identical protocol, lend confidence to the preparation of nucleotide-free Gα.

2.) *The authors mention a recent crystal structure between G-alpha and a 16-mer GEF peptide (pg. 4), that is similar in sequence as the one investigated in this study. In this light, what is the novel insight obtained in the present study compared to this earlier one? The NMR data show some CSPs in regions remote from the primary binding site, but an in depth structural characterization of this feature would require by far more effort than what has been presented here. And how reliable is the molecular modeling approach taken by the authors to visualize these potential conformational changes?*

Response to Comment #2- This comment from the reviewer has two parts. In the first part the reviewer wonders about the novelty of our findings in light of a previously reported crystal structure. In the second part the reviewer asks about the reliability of the molecular modeling approach to visualize the potential conformational changes induced by GIV.

PART 1: The novelty of our study compared to the earlier characterization of the crystal structure of G α in complex with a 16-mer peptide (KB-752) is two-fold. On one hand, our study provides insights into the structural basis for the binding of a naturally-occurring protein sequence with important biological functions (i.e., GBA motif), instead of for the binding of a non-native peptide (i.e., KB-752). On the other hand, our study shows that GIV binding causes perturbations in regions of the G protein that are very relevant for the mechanism of G protein activation (i.e., they are involved in nucleotide binding) while they were unchanged in the crystal structure with the synthetic peptide.

Thus, although the earlier crystal structure of a GIV-related peptide in complex with G α represents a valuable framework for our study, the findings reported in the current manuscript provide information that is biologically more relevant and that advances our understanding of the molecular mechanism of G protein activation. We have modified the text on Page 4 to clarify these points.

PART 2: Our molecular model does not permit to predict the conformational changes remote from the binding site because it is based on the crystal structure of G α i1 bound to a non-native peptide, which did not show such conformational changes. The goal of our modeling approach was to better characterize the protein-protein interface and to facilitate the visualization of what specific regions of the G protein distant to the protein-protein interface are altered upon GBA binding, rather than displaying the changes in conformation *per se*. This has been clarified on Pages 4 and 5, and by the addition of new panels to Supplementary Figure 2.

3.) *On pg. 8, the authors mention the interaction between a GPCR and G-alpha and discuss possible allosteric structural changes. A recent paper on G-alpha dynamics upon nucleotide and GPCR binding addresses this issue in detail and should be cited at this point (Goricancec et al, PNAS, 2016).*

Response to Comment #3- We agree with the reviewer and have included the reference in the point that s/he indicated as well as in the Introduction.

4.) *If the GBA motif in GIV is the main interaction site, what about the other 1100 aa in the protein? Is there any evidence of the relevance of this protein in mediating protein-protein interactions or regulation of its structure and interactions by PTMs? Is there any information on the domain structure of GIV?*

Response to Comment #4- We realize now that we had included a very limited amount of information on the domain structure of GIV and its protein-protein interactions. We have now included additional information on these by modifying the text (see Page 4) and including a scheme of the domain architecture of GIV in Supplementary Figure 2.

5.) *It seems surprising that binding of RGS proteins like the RGS-14 GOLOCO domain is not affected by disruption of the GIV protein interface on G-alpha. At least the binding groove on the G-alpha Ras domain is extremely similar to the GIV GBA binding site that has been identified here. Maybe a structural overlay of the Ga-GOLOCO and Ga-KB-752 structures can clarify this issue.*

Response to Comment #5- We would like to clarify first that it was not our intention to imply that binding of the GoLoco motif is not affected by disruption of the GIV binding interface on G α . Instead, what our data on Figure 4 shows is that some of the mutations that impair GIV binding also impair binding of the GoLoco motif. However, the overlap of the sets of G α mutations that affect binding to GBA or GoLoco motifs is only partial, and many of the overlapping mutations affect binding to both GBA and GoLoco motifs but to a significantly different extent.

The overlap of mutants that affect binding of both GBA and GoLoco motifs is likely explained by the observation of the reviewer, i.e., that they bind to a similar groove on G α . However, as suggested by the reviewer, comparison of the G α -GoLoco and G α -KB-752 structures (see Ref.38 in the manuscript) indicates that this groove is not identical. More specifically, the SwII region of G α adopts a different conformation in both structures, which changes the shape of the groove and the contacts made with the two different binding motifs. This explains the different effect of the same set of G α mutations on GBA or GoLoco binding.

Because the G α -GoLoco and G α -KB-752 structures have been overlaid and compared extensively in a previous work (see Ref.38 in the manuscript), we have not included it again in the current manuscript. Instead, we have modified the text to clarify the point discussed above referencing to the previously published comparison of the two structures (see Page 8)

6.) *Page 12, third paragraph: typo: "... binding pocket I our NMR ..." should be "... in our NMR ..." and on page 13, "conformed" should be "confirmed".*

Response to Comment #6- We apologize for these oversights, which have been corrected.

Reviewer #2

General comments: This manuscript provides an in depth structure-function analysis of GIV and DAPLE, two proteins of the GBA family that activate G proteins, but which are not G protein coupled receptors. There are only a handful of non-receptor activators of G proteins. Other examples include Ric-8A and Ric-8B in higher eukaryotes and Arr4 in yeast.

The approach includes sophisticated NMR, computational modeling, mutagenesis and protein biochemistry. Experiments are done with fragments (210 aa) of GIV rather than full length protein (1870 aa). This appears to be unavoidable and the authors argue persuasively that it is a valid alternative. New mutations are described that selectively uncouple Galpha from GIV or from the receptor, and these provide supporting evidence for the mode of action suggested by the NMR data. The mutations described have the potential to reveal the relative physiological significance of the two inputs to Galpha activation.

The overall conclusion, supported by the data, is that these GBA proteins share a common and specific site of binding to G proteins, and that there are fundamental differences in GBA-mediated and receptor-mediated G protein activation (Ric8 mediated activation is distinct from the other two mechanisms).

The paper is well written. The data are described appropriately, and interpreted conservatively. Overall I am enthusiastic about the submission.

Response to General Comments- We appreciate the positive comments from the reviewer and we are glad that s/he is enthusiastic about the manuscript. Below we provide a response for the concerns raised.

Concerns (in approximate order of importance):

1- The NMR analysis is the strength of the paper, but there is no comparable NMR data for Galpha when bound to GDP and receptor. There is a crystal structure of Galpha bound to a receptor, but it is static, contains two additional G protein subunits, and is in the nucleotide-free state. Thus it is somewhat of an apples-to-oranges comparison. This should be clarified in the text.

Response to Comment #1- We have modified the text on Page 10 to clarify this.

2- A 3D structural model of the GBA motif of GIV (residues 1678-1696) bound to Galpha is based on homology with the KB-752/Galalpha structure. The authors note sequence similarities between GIV and KB-752. Given the similar structures of Galpha-GDP plus or minus KB-752, how confident are these authors about the similarity on which the homology model is based? Is it feasible to do the NMR on Galpha plus KB-752, allowing a more valid comparison?

Response to Comment #2- This comment overlaps with a point raised by Reviewer #1 (see Comment#2). In our response to him/her (*Response to Comment#2, PART 2*) we provide an explanation of the purpose and limitations of our model. Briefly, we believe that the model is reliable for the protein-protein interface (see below for a justification) but it was not intended nor has the power to predict the nature of conformational changes distant to the protein-protein interface. We would like to clarify that G α -GDP plus or minus KB-752 are structurally very similar in the nucleotide

binding region but differ in the KB-752-binding site, which provides the basis for using it as a model to understand the protein-protein interface formed by the GIV GBA motif similar to KB-752.

We are confident about the similarity on which the model was built because of several lines of evidence presented in this manuscript and in previously published work. First, GIV and KB-752 compete for binding to $G\alpha$ (PMID: 19211784), suggesting a common binding site on $G\alpha$. Second, sequence similarity with the KB-752 and rudimentary modeling allowed us to identify residues in the GBA motif of GIV that are involved in binding to $G\alpha$ (PMID: 19211784, PMID: 22308453). Importantly, we could predict not only mutations that disrupt binding but also a mutation that increases it based on similarity with KB-752 (PMID: 22308453), further supporting that it is a valid template for modeling. Third, the biochemical data presented in the current manuscript from the analysis of >20 $G\alpha$ mutants and >150 mutant peptides of GIV's GBA motif are in very good agreement with the model. For these reasons, we believe that we generated a high confidence model and that an NMR analysis of $G\alpha$ bound to KB-752 would not provide significant additional information.

3- Mutations that uncouple $G\alpha$ from GIV or from the receptor are not tested in cells or *in vivo*. This is probably beyond the scope of the paper, but worth mentioning as a goal for the future.

Response to Comment #3- We believe that the reviewer refers to testing mutations in mammalian cells instead of in yeast cells as shown in Figure 7. Although we agree that it would be important to test mutants in mammalian cells and *in vivo*, there are some caveats for the implementation of such experiments. The main one is that the mutations might affect the regulation of $G\alpha$ by other regulators expressed in mammalian cells, which would preclude the unequivocal interpretation of results.

For example the GBA-binding deficient mutant N256E tested in Figure 7 also impairs significantly binding of GoLoco motifs (Figure 4). Thus, each mutant would require to be more extensively characterized to be able reach conclusions about the specific role of GBA-mediated regulation in the type of experiments mentioned by the reviewer. This is in part one of the reasons why we chose to test mutants in a system like yeast in which the G protein regulatory machinery is less complex (e.g., they do not express GoLoco motif-containing proteins). This has been further clarified in the text (see Page 10).

4- Should the peptide array data include a scrambled peptide negative control?

Response to Comment #4- We believe that showing such control is not necessary because there are many non-binding peptides within the same experimental set that serve as negative controls. Because each of these peptides differs only in a single amino acid, we believe that they are even more stringent controls than a scrambled sequence. In addition, we have performed many additional $G\alpha i3$ experiments with peptides arrays that support that the binding to the GBA motif of GIV is specific (e.g., $G\alpha i3$ does not bind to many peptides of the same length, even if they share sequence similarity with the GBA motif of GIV).

5- Use of the word "affinity" is unwarranted in places where binding is not quantified in any way (pull downs for example, vs. NMR binding analysis).

Response to Comment #5- We have revised the manuscript thoroughly for proper use of the term “affinity”.

6- Please defend the statement: “GBA proteins bind monomeric Galpha and do not require accessory proteins, as opposed to GPCRs, which require Gbeta/gamma.” – What is the relative exchange rate of Galpha alone coupled to a GPCR vs. GBA? What is the relative exchange rate of Galpha/beta/gamma coupled to a GPCR vs. GBA? If these data are unpublished (refs 30 and 32) they should be included or supported with published data instead.

Response to Comment #6- We realize now that our statement might have been an oversimplification. We have modified this statement, including references to support it, to express better the message that we were trying to convey (i.e., clarifying the preferred and direct substrate for the GEF activity of GIV or GPCRs).

The new statement reads: “First, GBA proteins displace Gβγ from the trimeric complex^{18,20} and exert their GEF activity directly on monomeric Gα^{18-20,25}, whereas GPCRs work most efficiently as GEFs when using intact Gαβγ trimers instead of monomeric Gα as substrate⁴⁰” (see on Page 12)

On a related note, it is difficult at this time to compare the relative efficiency of GBA and GPCRs *in vitro* due to differences in experimental systems (e.g., solution vs. membrane-reconstituted, different types of Gα and/or Gβγ). However, in a paper published while this manuscript was under review we have shown that GIV and GPCRs activate G proteins to a similar extent in cells by using BRET-based sensors of G protein activity. We have now discussed this point (see Page 11) and included the corresponding reference.

7- Please proof read the document carefully (“undergoes changes in the exact same regions”)

Response to Comment #7- We have revised the manuscript carefully to avoid mistakes.

Reviewer #3

General comments: The authors present a very complete characterization of the interaction between G-protein and GAB proteins. The combination of computational and experimental approaches allowed also to characterize the conformational changes and mechanisms underlying this interaction.

The manuscript is well written and the figures although they are a lot, are needed to comprehensively follow the article.

Response to General Comments- We appreciate the positive comments from the reviewer. Below we provide a response for the concerns raised.

Minor comments:

In the methods section, particularly the computational part need more details regarding the methods, programs and algorithms used. With the data that the authors are providing, I think it is hard to reproduce their calculations.

Response to Minor Comment- We have now revised the methods section to provide more detail. We have also deposited the coordinates of our model in a publicly available repository to facilitate the use and analysis of this model by other investigators.

REVIEWERS' COMMENTS:

Reviewer #1 (Remarks to the Author):

In their revised manuscript and the rebuttal letter, the authors now clarified my minor concerns. Therefore, I recommend publication of this paper in its present form.